# Primary visual cortex straightens natural video trajectories

Olivier J. Hénaff [1,5,7✉], Yoon Bai [2,6,7✉], Julie A. Charlton[2], Ian Nauhaus[2], Eero P. Simoncelli [1,3,4] & Robbe L. T. Goris [2✉]

Many sensory-driven behaviors rely on predictions about future states of the environment. Visual input typically evolves along complex temporal trajectories that are difficult to extrapolate. We test the hypothesis that spatial processing mechanisms in the early visual system facilitate prediction by constructing neural representations that follow straighter temporal trajectories. We recorded V1 population activity in anesthetized macaques while presenting static frames taken from brief video clips, and developed a procedure to measure the curvature of the associated neural population trajectory. We found that V1 populations straighten naturally occurring image sequences, but entangle artificial sequences that contain unnatural temporal transformations. We show that these effects arise in part from computational mechanisms that underlie the stimulus selectivity of V1 cells. Together, our findings reveal that the early visual system uses a set of specialized computations to build representations that can support prediction in the natural environment.

---

[1] Center for Neural Science, New York University, New York, NY, USA. [2] Center for Perceptual Systems, University of Texas at Austin, Austin, TX, USA. [3] Courant Institute of Mathematical Sciences, New York University, New York, NY, USA. [4] Flatiron Institute, Simons Foundation, New York, NY, USA. [5] Present address: DeepMind, London, UK. [6] Present address: Department of Brain and Cognitive Sciences, Massachusetts Institute of Technology, Cambridge, MA, USA. [7] These authors contributed equally: Olivier J. Hénaff, Yoon Bai. ✉email: henaff@google.com; yhb@mit.edu; Robbe.Goris@utexas.edu

The computations implemented by sensory systems are shaped by the properties of the physical environment and the behavioral demands of organisms. Many essential tasks such as catching prey and escaping predators rely on predictions about the future state of the world. Naturally occurring sensory input typically has a complex relationship with future environmental states, and it has been hypothesized that organisms transform these inputs into internal representations that facilitate temporal prediction[1–5]. Consider vision. Under natural circumstances, the patterns of light projected on the retina evolve according to nonlinear dynamics that are difficult to extrapolate. We hypothesize that the visual system transforms its input into a representation that follows a "straighter" trajectory through time (the temporal straightening hypothesis[6]). In this representation, prediction can be achieved through simple linear extrapolation.

Testing the temporal straightening hypothesis requires comparing the straightness (conversely, curvature) of the trajectory of a sequence of images in the pixel domain with that of the visual system's internal representation. We previously developed a method for estimating the curvature of perceptual representations of video sequences and used it to demonstrate that the human visual system selectively straightens natural videos[6]. Is this effect directly observable in the activity of neural populations, and if so, what are the response properties that underlie it? Here, we examine these questions in macaque primary visual cortex (V1). We used multi-electrode arrays to record V1 activity in anesthetized macaques while presenting frames taken from brief video clips. We developed a procedure for estimating the curvature of the neural population representation of these video sequences and compared this value to its pixel domain counterpart.

Our analysis revealed that the early visual system transforms the frames of natural videos so as to straighten their V1 population trajectories. This temporal straightening process is specific to natural sequences: the same analysis of neural responses to frames of synthetic videos that contain unnatural transformations

revealed that V1 populations substantially entangle such sequences. To gain insight into the computations underlying these effects, we examined the behavior of a stimulus response model built from operations of linear filtering, nonlinear pooling, and divisive normalization. This model captured the selectivity of individual V1 neurons for elementary stimulus attributes (orientation, scale, and phase), but only partially explained responses of single units to natural images. Despite this, the model largely replicated differences in population-level straightening effects across natural videos. We show that this behavior critically depends on the model's nonlinear pooling. Together, these findings establish that the early visual system processes its inputs using a set of specialized computations that can support perceptual prediction in the natural environment.

## Results

We compared the curvatures of brief video clips in the pixel and neural domains. In the pixel domain, each video frame corresponds to a point in a high-dimensional space whose coordinates are the brightness of individual pixels. A video clip corresponds to a sequence of such points. Consider three consecutive frames (Fig. 1a). A natural measure of local curvature is the unsigned angle between the segments that connect the middle frame to adjacent frames (Fig. 1b). We define the global curvature of a video clip as the average of these local curvatures. This measure, known as discrete curvature, is positive-valued and achieves a minimal value of zero only for straight sequences. The larger its value, the more curved a sequence is, and the more error-prone a linear extrapolation of the trajectory would be.

**Estimating neural curvature.** We recorded the activity of populations of up to a few dozen V1 neurons while presenting frames taken from brief video clips (see "Methods"). In our previous perceptual experiments, we used an experimental

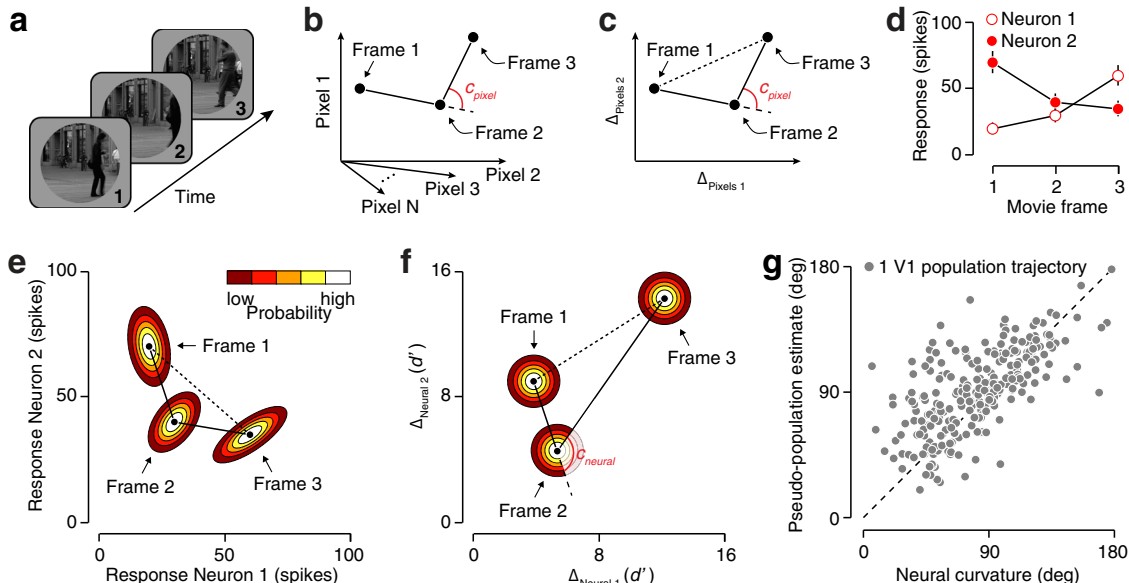

**Fig. 1 Measuring curvature in the pixel and neural domains. a** An example sequence of three movie frames. **b** Visualization of a high-dimensional representation of this sequence. Each point corresponds to a single movie frame, with each coordinate specifying the brightness of a pixel within that frame. Discrete curvature $c_{pixel}$ is defined as the unsigned angle between two segments connecting adjacent frames. **c** The curvature of a sequence is fully determined by the collection of pairwise distances. **d** Simulated responses of two neurons to a sequence of three movie frames. Error bars illustrate ± 1 standard deviation. **e** Joint response probabilities for these two neurons, for each of the three frames. **f** Representation of the same sequence of frames in a two-dimensional neural distance space. The distance metric is frame discriminability (in units of d'—Euclidian distance divided by standard deviation). **g** Comparison of neural curvature estimates obtained under a response model that includes correlations (abscissa) to one that assumes independence (ordinate).

protocol that relies on the presentation of static images, thereby isolating the contribution of spatial processing mechanisms[6]. Here, we seek to study the neural basis of these perceptual effects. We therefore used the same stimulus presentation method. Movie frames were presented in randomized order, interleaved with a blank screen. These manipulations removed any systematic contribution from fast motion-selective mechanisms, as well as from history-dependent mechanisms such as response adaptation. We then obtained neural response trajectories by arranging the data in the movies' natural temporal order.

The curvature of a trajectory is fully determined by the distances between all pairs of frames. Moreover, the pairwise distances between a set of $T+1$ points are uniquely constrained by their relative positions within a $T$-dimensional space (Fig. 1c). Mapping population representations into a neural distance space of this dimensionality (i.e., one less than the number of movie frames) is therefore sufficient for inferring neural curvature. Consider the responses of two neurons to a sequence of three movie frames (Fig. 1d). Their joint activity constitutes a population trajectory (Fig. 1e). Neural responses are noisy— repeated presentations of the same stimulus elicit different patterns of population activity[7,8]—and this noise is generally believed to limit perceptual discriminability. We therefore used frame discriminability as neural distance metric. Specifically, the overlap of the population response distributions elicited by each frame provides a principled measure of this statistic (Fig. 1e).

The number of measurements obtained in typical experiments is insufficient to describe the joint population response distribution, even for populations of moderate size. For this reason, we fit a descriptive response model to the observed population activity and computed frame discriminability under this fitted model. We then mapped the population responses for each frame into a neural distance space such that pairwise Euclidean distances were equal to the discriminability of the model-predicted response distributions (Fig. 1f). Finally, we obtained a curvature estimate by inferring a distribution of neural trajectories that were consistent with the data, and computed the value of curvature that was most likely under this distribution (see "Methods"). This procedure allowed us to obtain largely unbiased neural curvature estimates (Supplementary Fig. 1).

Neural trajectories are not only shaped by the mean pattern of neural activity, but also by the variability of this pattern. Variations in response strength across repeated presentations of the same stimulus are often correlated across neurons[9]. Such "noise correlations" can impact stimulus discriminability by altering the overlap of population response distributions[10]. For the two-neuron example, the impact of noise correlations is considerable. Inspection of the average responses suggests that the neural trajectory is modestly curved (Fig. 1e). Yet, in neural distance space, the trajectory is highly curved (Fig. 1f). Moreover, shared variability affects the distribution of plausible neural trajectories and can therefore affect our curvature estimates. We wondered whether a simpler estimation procedure that ignores these correlations would produce the same answer. We found this not to be the case: for many recorded trajectories, treating populations as pseudo-populations by ignoring noise correlations gave a dramatically different estimate of curvature. For some trajectories, this increased the estimate, but for others, it decreased the estimate (Fig. 1g)[11].

**Neural straightening of natural videos**. Consider an example recording (Fig. 2). Individual neurons respond in a selective manner to a randomly ordered sequence of movie frames (Fig. 2a). Neural response trajectories are obtained by arranging the data in the movies' natural order, as can be seen for three example units (Fig. 2b). We refer to the responses of an entire population to a single movie as a dataset. For each dataset, we fit population responses with a descriptive model in which spikes arise from a Poisson process that is subject to slow gain fluctuations[8,12,13] (black lines in Fig. 2b, see "Methods"). These gain fluctuations were shared across neurons, giving rise to correlated spiking activity across repeated trials. We evaluated the model's ability to explain the empirically observed response mean, variance, and covariance (Fig. 2c, Supplementary Fig. 2). If the model performed well, we proceeded to use it as a basis for estimating the curvature of the neural trajectory.

The projections of the pixel-domain and neural-domain trajectories of a six-frame video onto their corresponding first two principal components illustrate our main results (Fig. 2d). As can be seen, the neural trajectory is straighter than its pixel-

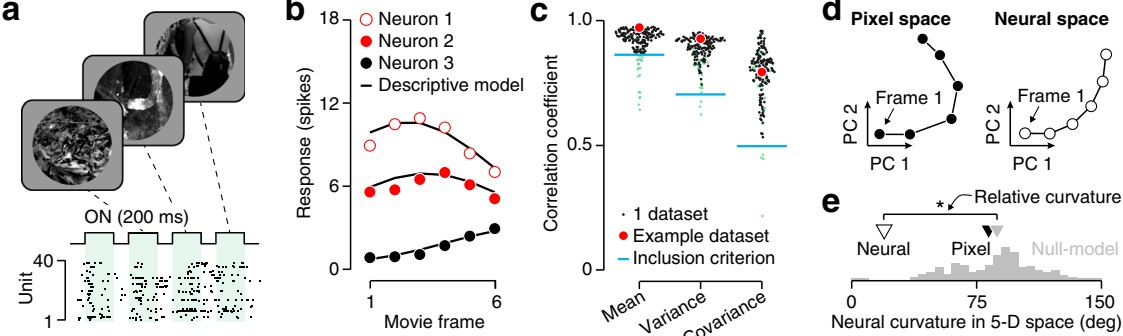

**Fig. 2 Testing the temporal straightening hypothesis. a** We presented individual frames from different video clips in randomized order. Each frame was shown for 200 ms, followed by 100 ms of constant luminance. We used multi-electrode arrays to record V1 population activity. The raster illustrates spiking activity recorded over a 1.3 sec period from an example V1 population ("Population 7", consisting of 39 units). **b** Mean response for three units in the example population, to frames of "Movie 7". We fit a descriptive response model to each unit. Points indicate mean spike counts, lines illustrate predicted responses of the fitted model. **c** We evaluated the model's goodness of fit by computing the correlation between predicted and measured response mean, variance, and covariance across the units and frames in each dataset. Each point corresponds to a dataset (red points indicate the example dataset). Blue lines indicate the inclusion criteria independently applied to these three statistics (included datasets are black points, excluded datasets are transparent green points). **d** Two-dimensional projections of the trajectory for the example dataset, in the pixel domain (left), and in the neural domain (right). **e** Estimated neural curvature of the example dataset (white triangle) and its expected distribution under the null model (gray histogram, see "Methods"). The black and gray triangle indicate the pixel-domain value and the null model's mean, respectively. Relative curvature is −69°, *P < 0.05, non-parametric test.

domain counterpart ($c_{pixel} = 34°$; $c_{neural} = 19°$). This difference is not due to dimensionality reduction: in their native 5-dimensional spaces the difference is even more substantial ($c_{pixel} = 83°$; $c_{neural} = 19°$). To evaluate the statistical significance of this effect, we computed the distribution of neural curvature estimates under a null model. We simulated a hypothetical population that was identical to the observed population in all regards, but whose neural trajectory preserved the pixel-domain

curvature (see "Methods"). For the example dataset of Fig. 2, the empirically observed curvature fell well beyond the central 95% interval of the null distribution (Fig. 2e). Neural curvature estimates can be biased, causing the average estimate under the null model to deviate from the pixel-domain value (Fig. 2e, black vs gray triangle). To take this potential bias into account, our analysis hereafter focuses on the difference between the empirical curvature estimate and the average estimate of the null model. We refer to this as "relative curvature". For the example dataset, its value is −69°.

V1 trajectories elicited by natural movies were typically straighter than their pixel-domain inputs (median relative curvature = −9.9°, $P < 0.001$, $n = 127$ datasets, Wilcoxon signed-rank test), but the effect varied substantially across datasets (Fig. 3a; 23 of 127 datasets were significantly straighter than the null model). This variability existed across neural populations (Fig. 3b; $P < 0.001$, $F_{1,123} = 11.87$, $n = 7$, analysis of variance), as well as across movies (Fig. 3c; $P < 0.01$, $F_{1,123} = 7.77$, $n = 20$). These effects did not interact statistically ($P = 0.28$, $F_{1,123} = 1.17$). Visual inspection of the movies revealed that straightening was generally stronger for movies containing dense textures rather than isolated objects (Fig. 3c inset, Supplementary Fig. 3). Finally, we verified that our conclusions were robust to the choice of null model, by comparing neural curvatures directly to pixel-domain curvature (Supplementary Fig. 4).

**Neural entangling of unnatural videos.** We have shown that V1 populations represent images in a manner that tends to straighten natural videos. This does not imply, however, that this representation is specifically tailored for this purpose. Following the approach of our perceptual experiments[6], our stimulus set therefore included twenty unnatural image sequences that fade from the first to the final frame of each of the natural videos (see Fig. 4a and Supplementary Fig. 3 for examples). Such sequences are unlikely to occur in the real world (e.g., the middle frame contains the average of the first and last frames—a "double exposure"). Perceptually, we found that their curvature typically increased[6], presumably because they are entangled by the nonlinear transformations of the early visual system[14].

We fit the response model to the population activity elicited by the unnatural movies (Fig. 4b, c). The projection of the pixel-domain and neural-domain trajectories of an unnatural movie onto

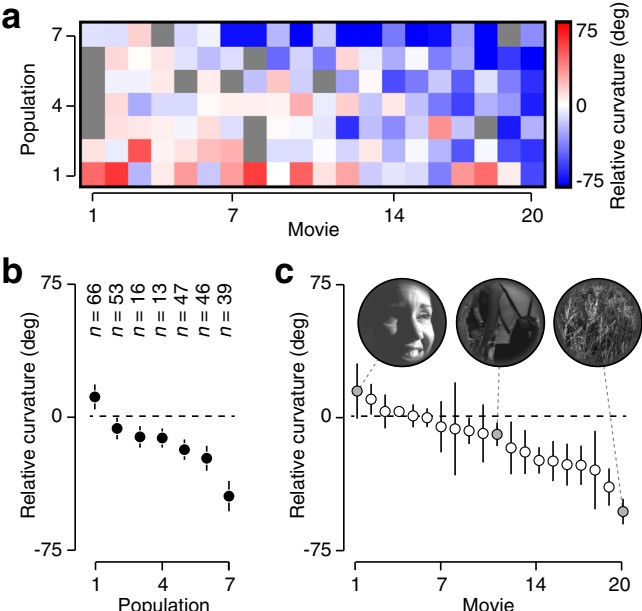

**Fig. 3 Curvature reduction for natural movie sequences. a** Relative curvature for seven V1 populations probed with twenty natural movies. Blue colors indicate neural straightening, red colors indicate neural entangling, and gray indicates missing data (see "Methods"). **b** Relative curvature for each V1 population, averaged across all movies. Error bars indicate s.e.m. across movies, $n$ values indicate population size. **c** Relative curvature for twenty movies, averaged across all populations. Error bars indicate s.e.m. across populations. Insets illustrate a single frame from three movies that elicited no, medium, and strong straightening (left, middle, and right, respectively; see "smile", "walking" and "prairie" in "Methods").

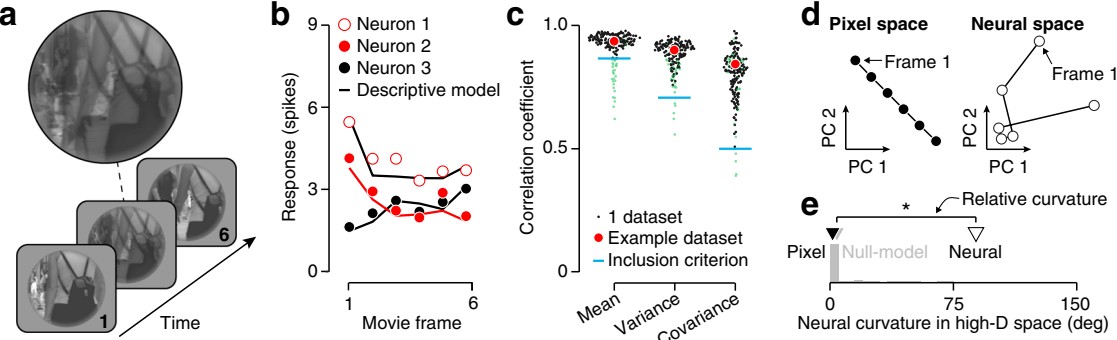

**Fig. 4 Testing the specificity of temporal straightening for natural sequences. a** An example sequence illustrating the first, middle, and last frame of an "unnatural" movie, constrained to be straight in the pixel domain. **b** Mean responses of three example units (Population 1) to frames of Movie 1. Points indicates mean spike counts, lines illustrate predicted responses of the descriptive response model. Prediction line for Neuron 2 is shown in red for clarity. **c** The correlation between predicted and measured response mean, variance, and covariance across all units and frames in each dataset (red points correspond to the example dataset). Blue lines indicate the inclusion criteria independently applied to these three statistics (included datasets are black points, excluded datasets are transparent green points). **d** Two-dimensional projections of trajectory of the example dataset, in the pixel domain (left), and in the neural domain (right). **e** Neural curvature of the example dataset (white triangle) and its expected distribution under the null model (gray histogram). The black and gray triangle indicate the pixel-domain value and the null model's mean, respectively. Relative curvature is 82°. *$P < 0.05$, non-parametric test.

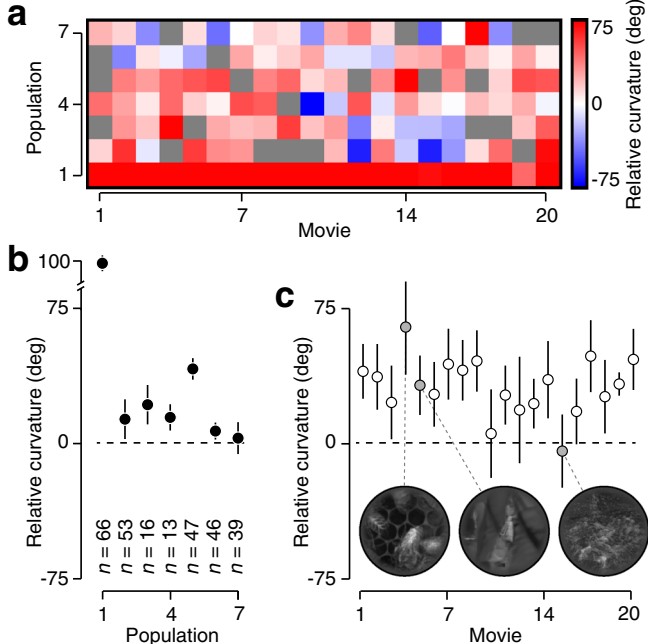

**Fig. 5 Curvature increase for unnatural sequences. a** Relative curvature for seven V1 populations probed with twenty unnatural movies. The ordering of the unnatural movies is matched to the ordering of the corresponding natural movies (Fig. 3). **b** Relative curvature for each V1 population, averaged across all movies. Error bars indicate s.e.m. across movies, *n* is population size. **c** Relative curvature for twenty movies, averaged across all populations. Error bars indicate s.e.m. across populations. Insets illustrate a single frame from three movies that elicited strong, mild, and no entangling (left, middle, and right, respectively; see "bees", "walking", and "carnegie dam" in "Methods").

the first two principal components reveal a very different outcome (Fig. 4d). As can be seen, the neural trajectory is much more curved than its pixel-domain counterpart ($c_{pixel} = 1°$; $c_{neural} = 80°$). This visual impression was confirmed by calculating the relative curvature in the full-dimensional neural response domain (Fig. 4e; mean $c_{null} = 6°$; $c_{neural} = 88°$, $P < 0.05$). V1 trajectories elicited by unnatural movies were typically more curved than their pixel-domain inputs (Fig. 5a; estimated curvature fell outside the central 95% interval of the null distribution for 25% of datasets; median relative curvature = 24.4°, $P < 0.001$, $n = 119$ datasets). There was modest variability in the entangling effect across neural populations (Fig. 5b; $P = 0.05$, $F_{1,115} = 3.79$, $n = 7$), though this effect was partly due to an outlier (population 1). There was no consistent difference across movies (Fig. 5c; $P = 0.32$, $F_{1,115} = 0.98$, $n = 20$). In summary, these control analyses suggest that temporal straightening by V1 populations is specific to image sequences that occur under natural conditions.

**Effects of spatial and temporal scale**. So far, we have presented evidence that V1 population representations straighten naturally occurring image sequences and distort unnatural ones. We hypothesize that the straightening or entangling of a sequence depends on its probability of occurring in the real world. If so, sequences which have equal real world probability should display similar amounts of straightening. Natural scenes are approximately scale-invariant, both spatially[15,16] and temporally[17]. Modest changes in either the spatial or temporal scale of a video should preserve its real world probability and it should therefore undergo a similar amount of straightening. Our experimental stimuli were constructed to test this prediction. Specifically, our

stimulus set of twenty natural and unnatural movies included ten unique movies that were each displayed at two spatial scales: *zoom × 1* (the original scale) and *zoom × 2* (created by upsampling the central portion of each frame—see Fig. 6a for an example). Likewise, for each 6-frame movie, we also included 5 intermediate frames, enabling us to compare two temporal scales (*frame rate × 1* and *frame rate × 2*—see Fig. 6b). Consistent with natural videos' statistical scale-invariance, we found that the pixel-domain curvature of these sequences was largely preserved across spatial and temporal scales ($r = 0.96$ across spatial scales, $r = 0.91$ across temporal scales; Supplementary Fig. 5). For natural movies, there was also a clear association of the neural representations across spatial scales: movies that elicited stronger straightening at the coarser scale tended to do so as well at the finer one (Fig. 6a; $r = 0.80$, $P < 0.01$, $n = 10$). Similarly, there was a strong association of relative curvature across temporal scales (Fig. 6b; $r = 0.71$, $P < 0.001$, $n = 20$ movies).

Unnatural sequences do not occur in the real world, and their entangling can therefore not be a design goal, but must instead be a byproduct of the non-linearity of the visual system. As such, we would not expect entangling to be a robust phenomenon. For example, two sequences that are displayed at different scales need not be equally entangled by neural transformations. Indeed, for the unnatural videos, we found no significant correlations in relative curvature across spatial (Fig. 6a; $r = 0.23$, $P = 0.53$, $n = 10$) and temporal scales (Fig. 6b; $r = 0.14$, $P = 0.56$, $n = 20$).

Note that on average, the *zoom × 1* condition elicited stronger straightening for natural movies than *zoom × 2* (Fig. 6a; $P < 0.05$, $n = 10$ movies, Wilcoxson signed-rank test of difference in relative curvature). Why might this be? V1 neurons are selective for the spatial scale of visual input[18]. The *zoom × 1* condition elicited stronger responses than *zoom × 2* at the level of populations (Fig. 6c, mean response histograms; $P = 0.01$, $n = 61$ paired datasets, two-sided Wilcoxson signed-rank test), and individual units ($P < 0.001$, $n = 2614$ units, two-sided Wilcoxson signed-rank test), indicating that those videos were better matched to the preferred spatial scale of the recorded populations. We therefore asked whether firing rate was associated with straightening, and found this to indeed be the case (Fig. 6c; $P < 0.001$, $F_{1,123} = 12.59$, ANCOVA). After controlling for this relationship, there was no effect of spatial scale on relative curvature ($P = 0.60$, $F_{1,123} = 0.28$). The relationship between straightening and mean firing could potentially be explained by movies containing cells' preferred image features also engaging additional non-linearities of V1 circuits[19] (Supplementary Fig. 6). This interpretation comes with an important implication. These stimuli activate millions of cortical neurons, and it is unlikely that our recordings include the most strongly driven cells. As such, our estimates of relative curvature might understate the temporal straightening occurring in the full cortical population.

In summary, the straightening of natural sequences appears to be an intrinsic property of each sequence, whereas the entangling of unnatural sequences is more erratic.

**Computational basis of neural straightening**. Which neural computations give rise to the perceptual straightening of natural videos? To explore this, we constructed a computational model for V1 neurons that is simple enough to be fit to data, yet powerful enough to describe basic feature selectivity and nonlinear response properties. This model has its roots in the work of Hubel and Wiesel[20], and incorporates elements introduced by many others[21-29]. Neural responses are described using a cascade of two linear-nonlinear (LN) operations. Visual stimuli are first processed by a set of oriented linear spatial filters that are identical in orientation and spatial frequency tuning but have different

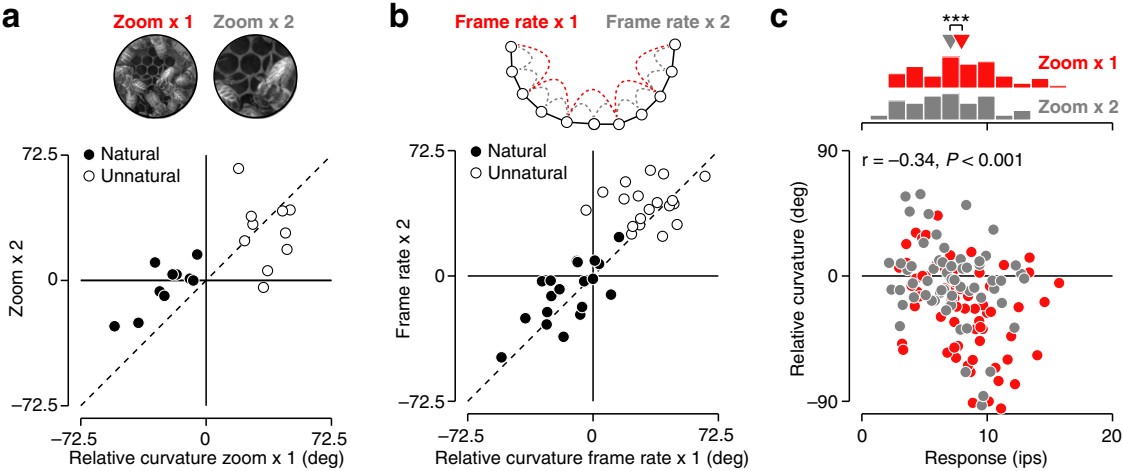

**Fig. 6 Comparing different spatial and temporal scales. a** Relative curvature for twenty movies, displayed at two spatial scales, averaged across all V1 populations. Black points indicate natural movies, white points unnatural ones (temporal scale is *framerate × 1*). **b** Relative curvature for forty movies, calculated for two temporal scales, averaged across all V1 populations. Black points indicate natural movies, white points unnatural ones (spatial scale is *zoom × 1*). **c** Relative curvature as a function of average firing rate for all natural movies. Each point illustrates a dataset, *zoom × 1* is shown in red, *zoom × 2* is shown in gray, temporal scale is *framerate × 1*. *P = 0.01, two-sided Wilcoxon signed-rank test, n = 61 datasets.

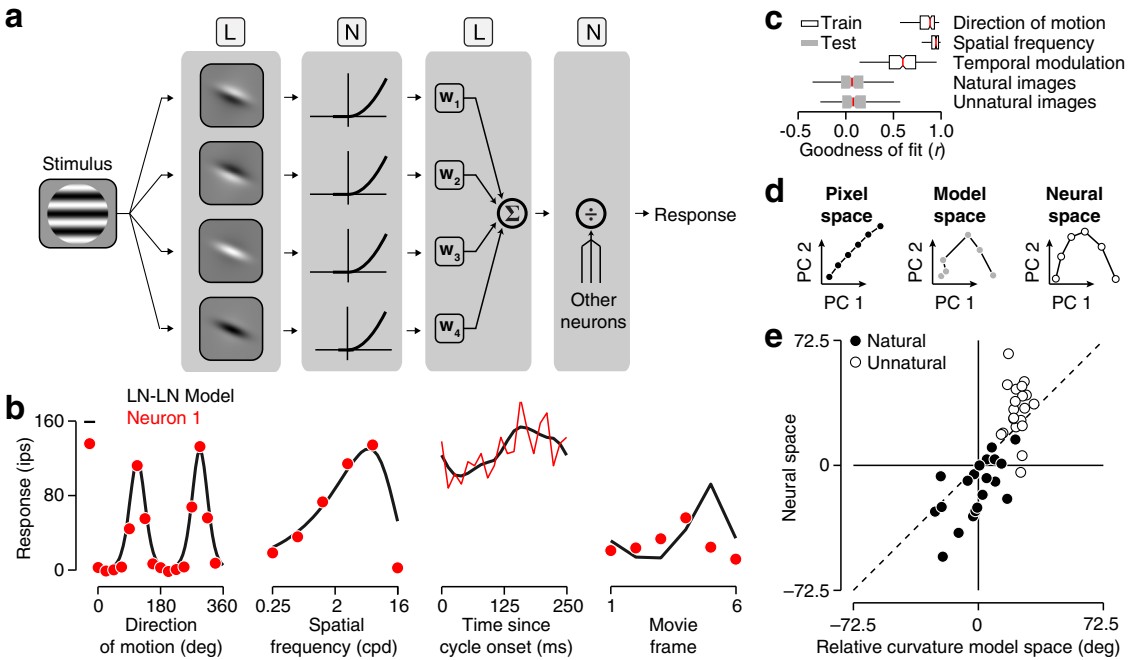

**Fig. 7 Relating V1 computation to temporal straightening. a** LN–LN model schematic. Stimuli are processed by a set of four linear Gaussian-derivative filters with phases differing by factors of 90°, followed by response exponentiation, linear pooling, and divisive normalization. **b** Measured and predicted tuning of an example neuron for direction of motion, spatial frequency, temporal modulation, and a natural movie ('Carnegie Dam', at *zoom × 1*). **c** Goodness-of-fit statistics across all individual V1 neurons. The model was "trained" on a set of white noise stimuli and a collection of drifting gratings. It was "tested" on the natural and unnatural movie frames. Box bounds represent the interquartile range, whiskers represent the range between the 1st and 99th percentiles. **d** Two-dimensional projections of an unnatural video's trajectory in the pixel domain (left), the model domain (middle) and in the neural domain (right). **e** Comparison of model-predicted and data-estimated relative curvature for forty movies, averaged across all V1 populations. Black points indicate natural movies, white points unnatural ones.

selectivity for spatial phase. The filter responses are nonlinearly transformed (halfwave rectified and squared), and then locally pooled. Finally, this pooled signal is subject to untuned divisive normalization to generate a response rate (Fig. 7a), which is used to generate spikes from a modulated Poisson process[8].

We probed all neurons within each V1 population with both white noise stimuli and drifting gratings (see "Methods").

Responses to the white noise stimuli enabled us to determine each neuron's receptive field location; responses to the drifting gratings revealed their selectivity for orientation, scale, and phase. For each neuron, we fit the parameters of the stimulus-response model to these data (see "Methods"). Consider an example neuron, for which the model provided a good account of basic stimulus selectivity (Fig. 7b). We quantified goodness of fit by

computing the correlation between the model's predicted responses and the measured tuning for orientation, spatial frequency, and temporal modulation while the other stimulus parameters were at the cells' preferred value. We found that the model typically performed well in describing these classic forms of stimulus selectivity (Fig. 7c; median $r = 0.89$, $0.95$, and $0.60$ for orientation, spatial frequency, and temporal modulation, respectively).

We also examined how well this model predicts individual cells' responses to natural and unnatural movie frames. In general, performance was much worse, at times close to chance (Fig. 7b, c; median $r = 0.07$, $P < 0.001$, $n = 440$ frames, Wilcoxson signed-rank test for natural frames; median $r = 0.08$, $P < 0.001$ for unnatural frames). These failures may indicate that the model lacks computational ingredients that are essential for explaining V1 activity in natural settings. If so, we might expect it to make poor predictions of the curvature of population trajectories. Alternatively, the model may fail to capture idiosyncratic response properties of individual neurons, but still provide a good summary of population trajectories.

For each dataset, we computed the population trajectory predicted by the stimulus-response model and compared its curvature to the empirically obtained estimate (see Fig. 7d for an example). Across all movies, there was a robust relationship between predicted and measured relative curvature (Fig. 7e; $r = 0.79$, $P < 0.001$, $n = 40$). Separated by movie type, this relationship remained substantial for natural movies ($r = 0.63$, $P = 0.003$, $n = 20$), but not for unnatural ones ($r = 0.02$, $P = 0.928$, $n = 20$). This is consistent with our results from previous analyses: the straightening of natural sequences displays a strong variation across videos (Fig. 3) and a strong association across spatial and temporal scales (Fig. 6). The entangling of unnatural sequences lacks both of these properties, indicating that it contains less explainable variability (the fraction of cross-movie variance is 26.6% for natural movies and 10.7% for unnatural movies).

Which elements of the stimulus-response model enable it to capture population-level straightening effects? To answer this question, we created restricted model versions by eliminating specific components. Removing untuned divisive normalization largely preserved the model's explanatory power (Fig. 8). This was also the case when we eliminated the simple cells (see "Methods", for reference: 60% of cells were "complex" when categorized by the relative temporal modulation of their firing rate to the preferred grating (F1/F0 < 1)[30]). On the other hand,

when we eliminated complex cells, the model's explanatory power was substantially diminished (Fig. 8). We conclude that, as with perceptual effects[6], the nonlinear pooling mechanism that underlies the phase-invariance of complex cells is largely responsible for the model's ability to explain patterns of straightening at the population level. This is broadly consistent with the view that complex cell pooling reduces temporal image fluctuations due to translation or other deformations[1,20,22,28,31–34].

While this LN–LN model provides insight into the computational basis of neural straightening, note that it systematically underestimated the values obtained from V1 measurements (median predicted relative curvature across all natural movies $= -0.6°$; median neural relative curvature $= -9.9°$) and failed to capture cross-population differences in curvature (natural movies: $r = -0.71$, $P = 0.077$, $n = 7$; unnatural movies: $r = 0.02$, $P = 0.962$, $n = 7$). We conclude that simple stimuli only partially engage the mechanisms that govern population responses to natural signals, and that additional model elements are necessary to fully explain the data.

## Discussion

What do sensory neurons seek to accomplish when they transform their inputs? We have tested the proposal that neurons in the visual system transform their inputs so as to straighten their temporal trajectories[6]. We developed a method to measure the curvature of a neural population trajectory and used it to analyze physiologically recorded V1 responses elicited by natural and unnatural image sequences. Our analysis revealed that the spatial processing mechanisms of the early visual system selectively straighten naturally occurring sequences.

Estimating the curvature of a neural population trajectory is a challenging statistical problem. The difficulty arises from several factors. First, neural responses are highly variable, leading to substantial variance in estimates. Second, realistic numbers of experimental trials and neurons are insufficient to yield precise estimates of joint response distributions or correlations. And third, curvature measurements in high-dimensional neural spaces are plagued by severe estimation bias (Supplementary Fig. 1a). We found that the combination of a principled statistical model of neural population activity, a probabilistic inference procedure for curvature, and an explicit null model of expected curvature estimates was sufficient to overcome these challenges (Supplementary Fig. 1b).

Our experimental test of the temporal straightening hypothesis only partially engages the mechanisms that shape neural trajectories under natural viewing conditions. In particular, by presenting static movie frames in a randomized order, interleaved by blanks, we excluded systematic contributions from history-dependent mechanisms such as temporal filtering, motion selectivity, response adaptation and recurrent computation, all of which are known to influence V1 responses. Despite this restriction, we found V1 representations to behave as predicted, consistent with a simple static model. How would these results generalize to representations of continuous streams of images? If temporal straightening is a prominent goal of visual processing, we might expect stronger effects under more natural stimulus conditions. Indeed, the inertia of physical systems makes their three-dimensional physical trajectories locally linear, and thus predictable. Although linear motion trajectories display a stable distribution of spatiotemporal energy, they can yield highly curved image-domain trajectories, since the brightness of individual pixels can change abruptly due to the passage of sharp edges, sudden occlusion, or changes in illumination. Engaging motion-selective neural mechanisms may therefore further

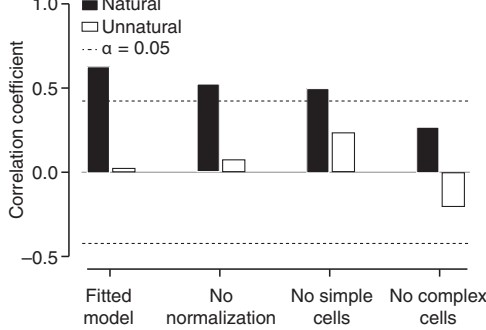

**Fig. 8 Dissection of the elements of the stimulus response model that shape neural trajectories.** The correlation between model-predicted and data-estimated curvature for different sub-models. From left to right: the full fitted model, model without divisive normalization, model with only complex cells, and model with only simple cells. Black bars indicate natural movies, white bars unnatural ones, the dotted line indicates a significance criterion of $P = 0.05$ (two-sided test).

increase temporal straightening. This is an important question for future work.

In recent years, the neuroscience community has witnessed a growing interest in characterizing the geometric properties of population representations. For example, recent studies of the motor cortex and the frontal cortex sought to measure the tangling[35], divergence[36], and curvature[37] of neural trajectories that unfold over hundreds of milliseconds. Each of these relied on analyses of trial-averaged trajectories and hence did not take response variability or correlation into account. While the simplicity of such analyses is appealing, we found that ignoring shared variability can dramatically alter neural curvature estimates in V1 (Fig. 1g). Our study is thus an example of how precise quantification of the geometry of neural trajectories may require the joint characterization of population activity[11]. In identifying the potential biases exhibited by population-level statistics, our results also highlight the need for an explicit null model when interpreting them[11]. The methodology we introduced for the estimation of neural curvature can be applied to other population datasets and generalized to other geometrical properties.

We also evaluated the predictions of a stimulus-response model that describes the primary selectivity properties of individual V1 neurons (in particular, for orientation, spatial frequency, and spatial phase) using linear filtering, the phase or position-invariance properties using nonlinear pooling, and the gain control properties using divisive normalization. Although this model did not accurately describe the responses of individual cells to natural videos[38], it was sufficient to account for the population-level pattern of straightening across natural videos. Comparison of different model parameterizations revealed that the nonlinear pooling mechanism played the most important role. How could the model be further improved? Under natural stimulation, intricate center-surround interactions shape the suppression at play in the primary visual cortex[39–41]. It seems likely that enriching our model to include these normalization mechanisms or those of earlier processing stages[6,42,43] could lead to a better account for our experimental findings.

We found that V1 population representations straighten natural videos, consistent with our previous perceptual findings[6]. But despite the use of largely identical stimuli, there was a substantial quantitative difference between these results (average straightening of ~10° in macaque V1, and ~30° in human perception), although the most responsive V1 populations did show effects nearly as large as the perceptual effect (Fig. 6c). One possible explanation for this discrepancy is that the best-driven subset of V1 neurons can fully explain the perceptual results, but that we typically failed to observe this subset because we did not tailor our stimuli to the recorded populations. Alternatively, it could be the case that downstream visual areas further straighten their inputs, and provide the substrate for perception. To examine this question, we have begun to assess the evolution of neural curvature along the visual hierarchy[44].

Finally, our results suggest that straightening might serve as an objective for learning more biologically plausible models of the visual system. Deep neural networks trained for object recognition are currently the best predictors of high-level visual function[45–47]. But these are typically trained on large sets of labeled images, and more stringent tests of their ability to account for human perception have revealed systematic deficiencies[43,48–50], including a failure to account for straightening[6]. These studies imply that a robust metric for the similarity of natural images is an important property of biological systems, and that this is currently lacking in artificial systems. An unsupervised objective that can learn to represent image structures and the natural continuous perturbation of

those structures as they evolve over time provides a promising alternative[1,4,6,33,51]. As such, temporal straightening could provide an effective principle underlying the emergence of visual representations that support the efficiency, flexibility, and robustness of biological intelligence.

## Methods

**Surgical preparation**. We recorded from four anesthetized, paralyzed, adult macaque monkeys (Macaca cynomolgus). Before surgery, each animal was initially anesthetized with ketamine (10 mg/kg, IM injection) and pretreated with 1.5 mg/kg diazepam and atropine sulfate (0.04 mg/kg, IM). Surgery consisted of the placement of cannulae in the saphenous veins of both legs as well as installation of a tracheal cannula. The animal's head was placed in a stereotaxic frame (David Kopf Instruments) and craniotomies and durotomies ($2 \times 2$ cm) were performed to expose the brain. Experiments typically lasted 5–6 days. Throughout the experiment, anesthesia was maintained with sufentanil citrate (4–20 µg/kg/h, IV), supplemented with isoflurane (0.5–2%) during surgeries. Animals were paralyzed using pancuronium bromide (0.1–0.2 mg/kg/h, IV). We monitored vital signs (heart rate, blood pressure, lung pressure, end-tidal $CO_2$, EEG, body temperature, urine flow and osmolarity), and maintained them within appropriate physiological ranges. Pupils were dilated with topical atropine. The eyes were protected with gas-permeable contact lenses and refracted with supplementary lenses chosen through retinoscopy. All procedures were approved by the University of Texas Institutional Animal Care and Use Committee and conformed to National Institutes of Health standards.

**Electrophysiology**. We recorded extracellular activity in area V1 using multi-shank electrode arrays (32 or 64 channels; Neuronexus A32, 2 or 4 shanks; A64, 8 shanks), advanced into the brain through a craniotomy and small durotomy. Once the electrode array was in place, we covered the exposed brain with a mixture of agarose and artificial cerebrospinal fluid. We allowed 30 min for the cortical tissue to settle before recording began.

**Visual stimulation**. We presented visual stimuli on a 20 inch gamma-corrected CRT monitor at a resolution of $1024 \times 768$ pixels with a refresh rate of 60 Hz. Visual stimuli were generated and presented using the Psychophysics Toolbox extensions for MatLab[52,53]. For each population, we first determined eye dominance, occluded the non-preferred eye, and estimated the population receptive field center. We then recorded responses to (1) sinusoidal gratings, (2) modified sparse-noise stimuli, and (3) static frames from natural and unnatural movie clips. Within each experiment, stimuli were centered on the population receptive field and presented in random order.

In the first experiment, we presented drifting sinusoidal gratings within a large circular aperture (diameter: 15 visual degrees). The stimulus set consisted of the combination of 16 equi-spaced directions of motion and 6 logarithmically spaced spatial frequencies (0.25–8 cycles/deg). Stimuli were presented for 1000 ms each, interleaved with a blank screen for 500 ms, and each was repeated 50 times.

In the second experiment, we presented modified sparse-noise stimuli, following the approach of ref. [54]. This stimulus set consisted of a collection of static bars ($2 \times 0.01$ degrees) with randomly chosen orientation, position, and polarity. Stimuli were presented for 133 ms each for approximately 20 min in total.

In the third experiment, we presented frames drawn randomly from 40 sequences. Ten of these were 11-frame clips extracted from movies: 'chironomus', 'bees', 'egomotion', 'prairie', 'carnegie dam', 'walking', 'water', and 'leaves-wind' (Chicago Motion Database, https://cmd.rcc.uchicago.edu), 'dogville' (a feature film, from Lions Gate Entertainment), and 'smile' (LIVE Video Quality Database[55,56]). All frames were achromatic and presented through a large aperture (diameter: 15 visual degrees). We generated rescaled versions of each original clip by extracting the central square of pixels, and upsampling and interpolating by a factor of two in both directions. We also designed artificial control sequences by linearly interpolating between the first and last frames of the corresponding natural sequence. In summary, 10 original movie clips produced 20 natural image sequences and 20 artificial image sequences. Single frames were presented for 200 ms each and interleaved with a blank screen for 100 ms. Each trial, all frames of all movies were presented exactly once, in randomized order. The experiment consisted of 50 trials in total.

**Extraction of neuronal response**. We used multi-electrode arrays to make extracellular recordings from V1. To extract responses of individual units, we first automatically spike-sorted these data with Kilosort2[57] (https://github.com/MouseLand/Kilosort2), followed by manual curation with the 'phy' user interface (https://github.com/kwikteam/phy). For each identified unit, we calculated its response by counting spikes in a time window whose duration matched that of the stimulus presentation. We chose a response latency for each unit by maximizing the stimulus-associated response variance[58].

**Measuring pixel-domain curvature**. Given an image sequence, we wish to compare its curvature in the domain of pixel-intensities with that of neural responses. Measuring the pixel domain curvature is straightforward. Let $\mathbf{x}_t^{\text{pixel}}$ be the vector of pixel intensities of the frame at time $t \in \{0, \ldots, T\}$. We define a sequence of normalized displacement vectors $\mathbf{v}_t^{\text{pixel}}$:

$$\mathbf{v}_t^{\text{pixel}} = \frac{\mathbf{x}_t^{\text{pixel}} - \mathbf{x}_{t-1}^{\text{pixel}}}{\left\| \mathbf{x}_t^{\text{pixel}} - \mathbf{x}_{t-1}^{\text{pixel}} \right\|} \tag{1}$$

and the curvature at time $t$ is simply the angle between two such vectors, which can be computed from their dot product:

$$c_t^{\text{pixel}} = \arccos\left( \mathbf{v}_t^{\text{pixel}} \cdot \mathbf{v}_{t+1}^{\text{pixel}} \right) \tag{2}$$

The global curvature of the sequence $c_{\text{pixel}}$ is the average of these local curvature values, in degrees.

**Modulated Poisson model of response distributions**. On a given trial $k$, the image $\mathbf{x}_t^{\text{pixel}}$ induces a pattern of recorded spiking activity which is summarized as a vector of spike counts $\mathbf{n}_t^k$, whose dimensionality $D$ is equal to the number of cells being recorded. We start by building a parametric model in which this spiking activity arises from a vector of spike rates $\boldsymbol{\lambda}_t$. Images are presented in random order, interleaved with a blank screen, and we assume that the spike counts in response to different images are independent from one another:

$$\mathbb{P}(\{\mathbf{n}_t^k\}|\{\boldsymbol{\lambda}_t\}) = \prod_{k=1}^{K} \prod_{t=0}^{T} \mathbb{P}(\mathbf{n}_t^k|\boldsymbol{\lambda}_t) \tag{3}$$

The simplest model of neural activity describes neural spike counts as distributed according to a Poisson distribution:

$$\mathbb{P}_{\text{Poisson}}(\mathbf{n}|\boldsymbol{\lambda}) = \prod_{i=1}^{D} \frac{\lambda_i^{n_i} \exp(-\lambda_i)}{n_i!} \tag{4}$$

We enrich this description by modeling spike-count correlations across cells, assuming their rates are modulated by a common gain factor[8,12]. Specifically, we assume they are independent and Poisson-distributed when conditioned on both the stimulus-driven rate, $\lambda_i$, and an unknown multiplicative gain $\mathbf{g}$. We model the gain as a multivariate log-normal random variable: $\mathbf{g} = \exp[\boldsymbol{\epsilon}]$, where $\boldsymbol{\epsilon} \sim \mathcal{N}(\boldsymbol{\mu}_{\boldsymbol{\epsilon}}, \boldsymbol{\Sigma}_{\boldsymbol{\epsilon}})$. The covariance matrix $\boldsymbol{\Sigma}_{\boldsymbol{\epsilon}}$ thus captures the dependencies across cells, and the mean vector $\boldsymbol{\mu}_{\boldsymbol{\epsilon}}$ is chosen such that the gain variable has unit mean (i.e. $\boldsymbol{\mu}_{\boldsymbol{\epsilon}} = -\frac{1}{2}\text{diag }\boldsymbol{\Sigma}_{\boldsymbol{\epsilon}}$, where diag extracts the diagonal of a matrix). The probability of a vector of spike counts is then:

$$\begin{aligned} \mathbb{P}(\mathbf{n}|\boldsymbol{\lambda}, \boldsymbol{\Sigma}_{\boldsymbol{\epsilon}}) &= \int d\mathbf{g}\, \mathbb{P}(\mathbf{g}|\boldsymbol{\Sigma}_{\boldsymbol{\epsilon}})\, \mathbb{P}(\mathbf{n}|\boldsymbol{\lambda}, \mathbf{g}) \\ &= \int d\mathbf{g}\, \mathbb{P}(\mathbf{g}|\boldsymbol{\Sigma}_{\boldsymbol{\epsilon}})\, \mathbb{P}_{\text{Poisson}}(\mathbf{n}|\boldsymbol{\lambda} \odot \mathbf{g}) \end{aligned} \tag{5}$$

where $\odot$ denotes the element-wise product between two vectors. We constrain $\boldsymbol{\Sigma}_{\boldsymbol{\epsilon}}$ to be the sum of a diagonal matrix (representing private gain modulation) and a low-rank matrix (representing shared gain modulation)[12]. We chose the rank of this matrix by cross-validation, resulting in the use of a rank-2 matrix throughout the analysis. The covariance matrix $\boldsymbol{\Sigma}_{\boldsymbol{\epsilon}}$ represents a stable property of the neural population, and we therefore share its parameters across all frames and videos.

**Summarizing responses with neural embeddings**. Having described the neural responses to each frame with a parametric distribution, we now summarize these distributions with a sequence of neural embeddings. We choose these embeddings such that their pairwise Euclidean distances are equal to the discriminability of the corresponding multi-dimensional response distributions. Specifically, we use the Fisher information metric $\mathcal{I}(\boldsymbol{\lambda})$ to approximate the discriminability of the rate vector $\boldsymbol{\lambda}$ from nearby vectors:

$$[\mathcal{I}(\boldsymbol{\lambda})]_{i,j} = \mathbb{E}_{\mathbf{n}}\left[ \frac{\partial}{\partial \lambda_i} \log \mathbb{P}(\mathbf{n}|\boldsymbol{\lambda}, \boldsymbol{\Sigma}_{\boldsymbol{\epsilon}}) \frac{\partial}{\partial \lambda_j} \log \mathbb{P}(\mathbf{n}|\boldsymbol{\lambda}, \boldsymbol{\Sigma}_{\boldsymbol{\epsilon}}) \right] \tag{6}$$

We simplify our analysis by first considering the Fisher information metric resulting from a modulated Poisson model with independent gain modulation for each neuron (the general case is described in the section "Numerical curvature calculation"). Under this approximation, the metric is diagonal with entries:

$$[\mathcal{I}(\boldsymbol{\lambda})]_{i,i} = \frac{1}{\lambda_i + \sigma_i^2 \lambda_i^2} \tag{7}$$

where $\sigma_i^2 = \exp([\boldsymbol{\Sigma}_{\boldsymbol{\epsilon}}]_{i,i}) - 1$. Thus, the discriminability of the rate vector $\boldsymbol{\lambda}$ is heterogeneous (i.e., it is a function of $\boldsymbol{\lambda}$).

We devised a monotonic nonlinear transformation that maps the rates into a space that is homogeneous in discriminability. Specifically, consider the following

transformation:

$$y_i = \frac{2}{\sigma_i} \sinh^{-1}(\sigma_i \sqrt{\lambda_i}), \tag{8}$$

which can be inverted to yield:

$$\lambda_i = \left[ \frac{1}{\sigma_i} \sinh\left( \frac{\sigma_i}{2} y_i \right) \right]^2. \tag{9}$$

The Fisher information metric for the transformed rate variables $y_i$ may be computed using the chain rule:

$$\begin{aligned} [\mathcal{I}(\mathbf{y})]_{i,i} &= \mathbb{E}_{\mathbf{n}}\left[ \frac{\partial}{\partial y_i} \log \mathbb{P}(\mathbf{n}|\mathbf{y}) \frac{\partial}{\partial y_i} \log \mathbb{P}(\mathbf{n}|\mathbf{y}) \right] \\ &= \frac{\partial \lambda_i}{\partial y_i} \mathbb{E}_{\mathbf{n}}\left[ \frac{\partial}{\partial \lambda_i} \log \mathbb{P}(\mathbf{n}|\boldsymbol{\lambda}) \frac{\partial}{\partial \lambda_i} \log \mathbb{P}(\mathbf{n}|\boldsymbol{\lambda}) \right] \frac{\partial \lambda_i}{\partial y_i} \\ &= \sqrt{\lambda_i + \sigma_i^2 \lambda_i^2}\, [\mathcal{I}(\boldsymbol{\lambda})]_{i,i}\, \sqrt{\lambda_i + \sigma_i^2 \lambda_i^2} \\ &= 1 \end{aligned} \tag{10}$$

Thus, the discriminability between response distributions for frames $t$ and $t'$ is simply the Euclidean distance between transformed rate vectors $\mathbf{y}_t$ and $\mathbf{y}_{t'}$. As such, the curvature of a neural trajectory may be estimated by computing the curvature of the transformed trajectory $\{\mathbf{y}_t\}_{t=0,\ldots,T}$ using the same procedure as in the pixel-domain.

**Variational Bayesian inference of neural curvature**. Next, we turn to the question of inferring the curvature of the sequence $\{\mathbf{y}_t\}_{t=0,\ldots,T}$ from data. It is tempting to solve this sequentially, by first estimating the parameters of each response distribution by maximizing their likelihood (Eq. (5)) given the dataset of all spike counts:

$$\{\hat{\boldsymbol{\lambda}}\}_t, \hat{\boldsymbol{\Sigma}}_{\boldsymbol{\epsilon}} = \arg \max_{\{\boldsymbol{\lambda}\}_t, \boldsymbol{\Sigma}_{\boldsymbol{\epsilon}}} \prod_{k=1}^{K} \prod_{t=0}^{T} \mathbb{P}(\mathbf{n}_t^k|\boldsymbol{\lambda}_t, \boldsymbol{\Sigma}_{\boldsymbol{\epsilon}}) \tag{11}$$

and then using these to estimate the most likely neural embedding $\{\hat{\mathbf{y}}_t\}_t$ (Eq. (8)), and taking the curvature of this. However this two-step procedure is plagued by substantial estimation bias when used with the amounts of data available in our experiments (Supplementary Fig. 1a). We therefore sought to compute the most likely curvature over a *distribution* of plausible neural embeddings.

Since a set of $T + 1$ points lie within a $T$-dimensional subspace, we start by parameterizing the $D$-dimensional rate vectors $\mathbf{y}_t$ as a function of $T$-dimensional vectors $\mathbf{x}_t^{\text{neural}}$ and an orthogonal embedding matrix $\mathbf{E}$:

$$\mathbf{y}_t = \mathbf{E}\mathbf{x}_t^{\text{neural}} \tag{12}$$

Since distances are preserved in this lower-dimensional subspace, so is the curvature of the sequence. We parameterize the trajectory in terms of the distances between successive points $d_t$, local curvatures $c_t$, and directions of curvature $\mathbf{a}_t$:

$$\begin{aligned} \mathbf{x}_t^{\text{neural}} &= \mathbf{x}_{t-1}^{\text{neural}} + d_t \mathbf{v}_t^{\text{neural}} \\ \mathbf{v}_t^{\text{neural}} &= \cos(c_t) \mathbf{v}_{t-1}^{\text{neural}} + \sin(c_t) \mathbf{a}_t \end{aligned} \tag{13}$$

These are themselves parameterized in terms of real-valued variables, on which we place the following priors:

$$\begin{aligned} d_t &= f_d(z_t^d) & z_t^d &\sim \mathcal{N}(f_d^{-1}(d^*), \sigma_d^2) \\ c_t &= z_t^c & z_t^c &\sim \mathcal{N}(c^*, \sigma_c^2) \\ \mathbf{a}_t &= f_{\mathbf{a}_t}(\mathbf{z}^{\mathbf{a}_t}) & \mathbf{z}^{\mathbf{a}_t} &\sim \mathcal{N}(\mathbf{0}, \boldsymbol{\Sigma}_{\mathbf{a}}) \\ \mathbf{E} &= f_{\mathbf{E}}(\mathbf{Z}^{\mathbf{E}}) & \mathbf{Z}^{\mathbf{E}} &\sim \mathcal{N}(\mathbf{0}, \mathbf{I}) \end{aligned} \tag{14}$$

where $f_d$ is a smooth rectifying function, $f_{\mathbf{a}_t}$ ensures that $\mathbf{a}_t$ is of unit length and orthogonal to $\mathbf{v}_{t-1}^{\text{neural}}$, and $\boldsymbol{\Sigma}_{\mathbf{a}}$ controls the effective dimensionality and aspect-ratio of the trajectory. The function $f_{\mathbf{E}}$ implements the Graham-Schmidt algorithm, which ensures that the columns of $\mathbf{E}$ are an orthonormal family of vectors in the $D$-dimensional neural space. This guarantees that the metric properties of the trajectory $\{\mathbf{y}_t\}_t$ (i.e. all pairwise distances, path length, and curvature) are identical to those of the trajectory $\{\mathbf{x}_t^{\text{neural}}\}_t$. Note that the optimal global curvature $c_{\text{neural}} = c^*$ given a sequence of local curvature values $\{c_t\}_t$ is simply their average, in keeping with our previous definition.

Let $\boldsymbol{\theta} = \{d^*, c^*, \sigma_d, \sigma_c, \boldsymbol{\Sigma}_{\mathbf{a}}\}$ be the parameters governing the global properties of the neural trajectory, and $\mathbf{z} = \{z_t^d, z_t^c, \mathbf{z}^{\mathbf{a}_t}, \mathbf{Z}^{\mathbf{E}}\}$ be those specifying its local properties. We wish to estimate the most likely value of the global parameters (including the global curvature) given the dataset of all spike counts $\mathbf{N} = \{\mathbf{n}_t^k\}_{t=0,T}^{k=1,K}$:

$$\begin{aligned} \boldsymbol{\theta}^* &= \arg \max_{\boldsymbol{\theta}} \log \mathbb{P}(\mathbf{N}|\boldsymbol{\theta}) \\ &= \arg \max_{\boldsymbol{\theta}} \log \int \mathbb{P}(\mathbf{N}|\mathbf{z})\mathbb{P}(\mathbf{z}|\boldsymbol{\theta})d\mathbf{z} \end{aligned} \tag{15}$$

Intuitively, these are the global parameters that are most consistent with the family of neural trajectories $\mathbf{z}$ that are supported by the data. However, this quantity is

impossible to compute in practice, given the dimensionality of $\mathbf{z}$. We therefore derive a lower bound on the likelihood[59], using an approximate posterior distribution over local variables $\mathbb{P}(\mathbf{z}|\mathbf{N}, \boldsymbol{\phi})$:

$$
\begin{aligned}
\log \mathbb{P}(\mathbf{N}|\boldsymbol{\theta}) &= \log \int \frac{\mathbb{P}(\mathbf{z}|\mathbf{N}, \boldsymbol{\phi})}{\mathbb{P}(\mathbf{z}|\mathbf{N}, \boldsymbol{\phi})} \mathbb{P}(\mathbf{N}|\mathbf{z})\mathbb{P}(\mathbf{z}|\boldsymbol{\theta}) d\mathbf{z} \\
&= \log \mathbb{E}_{\mathbf{z}|\mathbf{N},\boldsymbol{\phi}} \left[ \frac{\mathbb{P}(\mathbf{N}|\mathbf{z})\mathbb{P}(\mathbf{z}|\boldsymbol{\theta})}{\mathbb{P}(\mathbf{z}|\mathbf{N}, \boldsymbol{\phi})} \right] \\
&\geq \mathbb{E}_{\mathbf{z}|\mathbf{N},\boldsymbol{\phi}} \left[ \log \frac{\mathbb{P}(\mathbf{N}|\mathbf{z})\mathbb{P}(z|\boldsymbol{\theta})}{\mathbb{P}(\mathbf{z}|\mathbf{N}, \boldsymbol{\phi})} \right] \\
&\geq \mathbb{E}_{\mathbf{z}|\mathbf{N},\boldsymbol{\phi}}[\log \mathbb{P}(\mathbf{N}|\mathbf{z})] - D_{KL}[\mathbb{P}(\mathbf{z}|\mathbf{N}, \boldsymbol{\phi})||\mathbb{P}(\mathbf{z}|\boldsymbol{\theta})]
\end{aligned}
\tag{16}
$$

We chose the approximate posterior $\mathbb{P}(\mathbf{z}|\mathbf{N}, \boldsymbol{\phi})$ to be a Gaussian $\mathcal{N}(\boldsymbol{\mu}_{\boldsymbol{\phi}}, \boldsymbol{\Sigma}_{\boldsymbol{\phi}})$ with a diagonal covariance. Since the prior $\mathbb{P}(\mathbf{z}|\boldsymbol{\theta})$ is also Gaussian, the second term may be expressed in closed form. We use a Monto Carlo approximation of the first term[60], by sampling from the approximate posterior and evaluating Eq. (5). We maximize this lower bound with respect to the parameters of the prior and approximate posterior $(\boldsymbol{\theta}, \boldsymbol{\phi})$ using the Adam optimizer[61] as implemented by the PyTorch machine-learning library.

We use the resulting curvature estimates (which we refer to as "population-based curvature estimates") in subsequent analyses, unless mentioned otherwise. In order to assess the importance of including noise correlations in our estimate of the distribution of plausible neural trajectories, we repeated this analysis while constraining the gain modulation to be independent across neurons (i.e. $\boldsymbol{\Sigma}_{\boldsymbol{\epsilon}}$ is diagonal). We refer to these as "pseudo-population curvature estimates", and plot them on the $y$-axis of Fig. 1g and Supplementary Fig. 7a.

Our inference procedure also provides estimates of the average neural distance between successive frames $d^*$. We found this distance to correlate well with our ability to recover neural curvature. In particular, very small trajectories (e.g. $d^* < 0.25$) yielded unreliable curvature estimates, and we therefore excluded them from further analysis.

**Evaluating the model fit**. Having optimized the objective described above, we obtain a distribution over plausible neural trajectories, represented by the vector of parameters $\boldsymbol{\phi}$. In particular, the mean vector $\boldsymbol{\mu}_{\boldsymbol{\phi}}$ represents a trajectory which best captures the distributions over local distances and curvatures. We use this trajectory to evaluate the goodness of fit of our model. We use Eqs. (14) (left column), (13), (12), and (9) to compute a trajectory of rate vectors $\{\boldsymbol{\lambda}_t\}_t$. We compare these model-predicted rates to the trial average $\hat{\boldsymbol{\lambda}}_t = \frac{1}{K}\Sigma_{k=1}^{K}\mathbf{n}_t^k$, in the first column of Figs. 2c, 4c. Specifically, we compute the Pearson correlation $r$ between log-transformed rate distributions, and exclude from our analysis any trajectory whose $r^2$ was below 0.75.

To estimate the model-predicted spike-count variance and covariance, we start by computing the model-predicted gain covariance. Given that we enforce the gain to have unit-mean, this covariance is $\boldsymbol{\Sigma}_{\mathbf{g}} = \exp[\boldsymbol{\Sigma}_{\boldsymbol{\epsilon}}] - 1$, where the exponential is computed element-wise. Under the modulated Poisson model[8] the spike-count covariance is

$$
\boldsymbol{\Sigma}_t = \mathrm{diag}[\boldsymbol{\lambda}_t] + \boldsymbol{\Sigma}_{\mathbf{g}} \odot \boldsymbol{\lambda}_t \boldsymbol{\lambda}_t^{\top} \tag{17}
$$

where $\mathrm{diag}[\,\cdot\,]$ forms a diagonal matrix from a vector, $\odot$ denotes element-wise multiplication, and $^{\top}$ transposition. We compare these model-predicted covariances to the empirical covariance $\hat{\boldsymbol{\Sigma}}_t = \frac{1}{K-1}\Sigma_{k=1}^{K}(\mathbf{n}_t^k - \hat{\boldsymbol{\lambda}}_t)(\mathbf{n}_t^k - \hat{\boldsymbol{\lambda}}_t)^{\top}$ in the second and third columns of Figs. 2c, 4c. Here too we compute the Pearson correlation $r$ between predicted and observed log-transformed variances, and exclude any trajectory whose $r^2$ was below 0.5. We proceed similarly for the non-diagonal covariance entries, excluding negative entries and using an inclusion criterion of 0.25. We verified that our use of an inclusion criterion did not affect our conclusions (Supplementary Fig. 8).

**Numerical curvature calculation**. To define and estimate neural curvature, we sought to transform rate trajectories such that their pairwise Euclidean distances would represent the discriminability of the underlying response distributions. In the previous section, we derived this transformation under the assumption that gain modulation is independent across neurons (Eq. (8)). Our response model estimates the coupling of gain variability across neurons, hence we can assess whether this assumption has affected our estimates of neural curvature.

Given that the curvature of a sequence is fully determined by its pairwise distances, we seek to estimate the pairwise discriminability of our inferred response distributions, while taking into account the noise correlations induced by shared gain variability. The discriminability of two nearby distributions $t$ and $t'$ can again be computed using Fisher information:

$$
\left[d_F(\boldsymbol{\lambda}_t, \boldsymbol{\lambda}_{t'})\right]^2 = (\boldsymbol{\lambda}_t - \boldsymbol{\lambda}_{t'})^{\top} \left[\frac{\boldsymbol{\Sigma}_t + \boldsymbol{\Sigma}_{t'}}{2}\right]^{-1} (\boldsymbol{\lambda}_t - \boldsymbol{\lambda}_{t'}) \tag{18}
$$

where the covariance matrices $\boldsymbol{\Sigma}_t$ and $\boldsymbol{\Sigma}_{t'}$ are fully determined by their associated rate vectors (Eq. (17)).

For arbitrary, potentially distant, rate vectors this measure of local discriminability may no longer be valid. For these, we define a sequence of rate vectors $\{\mathbf{l}_i\}_{i \in [0,I]}$ connecting the two (i.e. $\mathbf{l}_0 = \boldsymbol{\lambda}_t$ and $\mathbf{l}_I = \boldsymbol{\lambda}_{t'}$). Given a

sufficiently dense sampling, we can compute the discriminability of successive points along the path $d_F(\mathbf{l}_i, \mathbf{l}_{i+1})$. Summing these local distances gives us the path length, or discriminability of the end-points while following the path $\{\mathbf{l}_i\}$. Finally, the geodesic distance between the end-points is the minimum of such path lengths:

$$
d_G(\boldsymbol{\lambda}_t, \boldsymbol{\lambda}_{t'}) = \min_{\{\mathbf{l}_i\}} \sum_{i=0}^{I-1} d_F(\mathbf{l}_i, \mathbf{l}_{i+1}) \tag{19}
$$

Following ref. [49] we compute the geodesic by iteratively adjusting a candidate sequence until it minimizes this objective. In this way we compute the geodesic distance between all successive rate vectors $d_G(\boldsymbol{\lambda}_t, \boldsymbol{\lambda}_{t+1})$ as well as rate vectors once removed $d_G(\boldsymbol{\lambda}_t, \boldsymbol{\lambda}_{t+2})$. We then use Al-Kashi's theorem (the law of cosines) to convert these distances into an estimate of curvature:

$$
\cos \gamma_t = \frac{d_G^2(\boldsymbol{\lambda}_{t-1}, \boldsymbol{\lambda}_t) + d_G^2(\boldsymbol{\lambda}_t, \boldsymbol{\lambda}_{t+1}) - d_G^2(\boldsymbol{\lambda}_{t-1}, \boldsymbol{\lambda}_{t+1})}{2 \cdot d_G(\boldsymbol{\lambda}_{t-1}, \boldsymbol{\lambda}_t) \cdot d_G(\boldsymbol{\lambda}_t, \boldsymbol{\lambda}_{t+1})} \tag{20}
$$
$$
c_t = \pi - \gamma_t
$$

We refer to the resulting values as "numerical curvature estimates". Note that these estimates are identical to the population-based estimates described previously if we approximate the covariance matrices $\boldsymbol{\Sigma}_t$ by their diagonal during the numerical calculation. We then assess the impact of this approximation by comparing our numerical curvature estimates which utilize the diagonal approximation and those that do not. Across the datasets we considered, we find this approximation to have a negligible impact (Supplementary Fig. 7c). As such the analytical rate-transformation we derived (Eq. (8)) provides a simple, accurate, and efficient way of converting rate trajectories into a geometrically interpretable space. We therefore use the resulting curvature estimates for all of our analyses, except when assessing the impact of noise correlations on curvature estimates (Fig. 1g and Supplementary Fig. 7 which use the numerical estimator on the $x$-axis).

**Fraction of cross-movie variance**. We estimated the fraction of variance in relative curvature that arises from variations in the stimulus. As is standard in ANOVA, we partitioned the total sum of squares into components arising from variations in movies ($SS_{\mathrm{movie}}$) and populations ($SS_{\mathrm{population}}$):

$$
\begin{aligned}
\sum_k \left(c_k - \bar{c}\right)^2 &= \sum_k \left(\overline{c_k} - \bar{c}\right)^2 + \sum_k \left(c_k - \overline{c_k}\right)^2 \\
&= SS_{\mathrm{movie}} + SS_{\mathrm{population}}
\end{aligned}
\tag{21}
$$

where $c_k$ is the relative curvature of the $k$th trajectory, $\bar{c}$ is the relative curvature averaged over all trajectories, and $\overline{c_k}$ is the relative curvature averaged over those trajectories in which the presented movie was the same as that of the $k$th trajectory. The fraction of cross-movie variance is given by: $SS_{\mathrm{movie}}/(SS_{\mathrm{movie}} + SS_{\mathrm{population}})$. We calculated this number for natural and unnatural movies separately.

**LN–LN model**. We fit the parameters of a two-stage feedforward model by analyzing responses from individual V1 neurons to sinusoidal gratings (experiment 1) and noise stimuli (experiment 2), and then used this model to predict each neuron's responses to complex natural and unnatural movie frames (experiment 3). The model describes how a static image is transformed into the firing rate of a cortical cell. In the first stage, stimuli are processed in four parallel channels, each composed of a linear spatial filter whose output is halfwave rectified. The filters have identical selectivity for orientation and spatial frequency, but differ in their phase selectivity, with the first filter preferring a phase of $\phi$ degrees, and the others preferring $\phi + 90$, 180, and 270 degrees, respectively. The spatial profile of each filter is given by the derivative of a 2-D Gaussian function (specific examples shown in Fig. 7a)[29]. At the preferred orientation, the spatial frequency selectivity of the filter solely depends on the order of the derivative $b$:

$$
r_{\omega}(\omega; \omega_0, b) \propto \left[ (\omega/\omega_0) e^{-\frac{1}{2}(\omega/\omega_0)^2} \right]^b \tag{22}
$$

where $\omega$ is stimulus spatial frequency and $\omega_0$ the filter's preferred spatial frequency.

The orientation selectivity of this filter depends on the aspect ratio of the Gaussian $\alpha$, the order of the derivative $b$, and the directional selectivity $d$:

$$
\begin{aligned}
r_{\theta}(\theta; \theta_0, \alpha, b, d) \propto \quad &\left[1 + \frac{d}{2}\left(\mathrm{sgn}\left(\cos(\theta - \theta_0)\right) - 1\right)\right] \cdot \\
&\left[\cos(\theta - \theta_0) \cdot e^{-\frac{1}{2}(1-\alpha^2)\cos^2(\theta-\theta_0)}\right]^b
\end{aligned}
\tag{23}
$$

where $\theta$ is stimulus orientation, $\theta_0$ the filter's preferred orientation, and parameter $d \in [0, 1]$ determines the direction selectivity. The function $\mathrm{sgn}(.)$ computes the sign of its argument (returning $-1$ for negative and $+1$ for positive), and the initial parenthesized expression serves to multiply the half of the tuning curve in the non-preferred direction by $(1 - d)$.

The second stage of the model consists of a linear combination of the four channel responses, followed by divisive normalization. The normalized response is transformed into a firing rate by including a spontaneous discharge

and a response scalar:

$$R = \varepsilon + \beta \left[ \frac{\sum_{i=1}^{4} w_i \max(0, L_i)^2}{\sigma^2 + \sum_{i=1}^{n} L_i^2} \right] \qquad (24)$$

where $\varepsilon$ is spontaneous discharge rate, $\beta$ controls the response range, $w_i$ is a pooling weight, and $L_i$ a linear filter's response. The normalization signal consists of the sum of a stimulus-independent normalization constant $\sigma$ and the pooled activity of a diverse set of neurons. We set $\sigma = 0.15$, and approximated the latter term with the root mean square contrast of the stimulus.

In total, the LN–LN model has 11 free parameters: five filter parameters (preferred orientation $\theta_0$, preferred spatial frequency $\omega_0$ and phase $\phi$, aspect ratio $\alpha$, derivative order $b$, and direction selectivity $d$), four pooling weights ($w_i$, constrained to sum to one, thus yielding three free parameters), and two parameters controlling response range and amplitude (spontaneous discharge rate $\varepsilon$ and response scale $\beta$). For each cell, we first optimized the filter and response range parameters by maximizing the likelihood over the observed grating responses calculated by counting spikes in a time window whose duration matched that of the stimulus presentation (1000 ms). We obtained a likelihood by assuming that spike counts arise from a modulated Poisson process[8]. We used a simplex algorithm and multi-start fitting procedure with semi-randomized starting values to find the best fitting parameters. Next, we estimated the four pooling weights $w_i$ by fitting the temporal response modulation exhibited during stimulation with the most effective grating (see Fig. 7b for an example). We estimated the temporal evolution of the firing rate by counting spikes in 12.5 ms time windows, and used least-squares regression to find the best fitting pooling weights.

We analyzed the responses to the modified sparse-noise stimuli shown in experiment 2 to obtain an estimate of the spatial location of each cell's receptive field center. Following the approach of ref. [54], we fit a 2-D Gaussian to the spatial response profile measured for black and white bars whose orientation approximated the cell's preferred value. We used the center of the Gaussian as an estimate of the cell's receptive field center.

Finally, we combined all of these parameter estimates to generate model predictions for the natural and unnatural stimuli of experiment 3. For each cell, we generated the appropriate quadrature set of spatial Gaussian-derivative filters, centered on the cell's receptive field center estimate. We obtained filter responses $L_i$ by cross-correlating these Gaussian derivatives with the stimuli, and then transformed these responses into a firing rate prediction using Eq. (24). For sinusoidal gratings, this image-computable version of the LN–LN model produced responses identical to those of the Fourier domain formulation.

To facilitate the comparison with our neural curvature estimates, we transformed the model-predicted firing rates with the same non-linearity derived from our descriptive model (Eq. (8)). We then computed the curvature of the transformed trajectory using the same definition as in the pixel domain. We evaluated the model's performance by computing the correlation between the predicted and observed curvature for each natural and unnatural movie, averaged across all populations (Fig. 7e).

**Dissection of LN–LN model**. To interrogate the role of each of the model's elements, we created multiple versions of the model by manipulating its parameter values (Fig. 8). Specifically, we created a version without divisive normalization by adopting all the parameter estimates of the fitted model, but setting the denominator in Eq. (24) to one. We also created a version in which all cells were turned into phase-invariant complex cells by setting all the pooling weights $w_i$ to 0.25. Finally, we created a version in which all cells were turned into simple cells by setting each cell's highest pooling weight $w_i$ to one and the three other weights to zero.

**Reporting summary**. Further information on research design is available in the Nature Research Reporting Summary linked to this article.

## Data availability
The data that support the findings of this study are available in the following repository: https://doi.org/10.17605/OSF.IO/VF2XK. Source data are provided with this paper.

## Code availability
The analysis code that supports the findings of this study is available from the corresponding authors upon reasonable request.

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

## Acknowledgements
The authors thank Corey Ziemba and Zoe Boundy-Singer for help with data-collection and valuable discussions. This work was supported by UT Austin (startup funds provided to RLTG and IN), and the Howard Hughes Medical Institute (Investigatorship to EPS).

## Author contributions
O.J.H., Y.H.B, E.P.S., and R.L.T.G. conceived and designed the study. I.N. developed the physiology setup and performed experimental preparations. Y.H.B., J.A.C., and R.L.T.G. collected data. O.J.H. and E.P.S. developed the curvature estimation procedure. O.J.H. and Y.H.B. analyzed data. O.J.H. and R.L.T.G. wrote the manuscript with input from Y.H.B. and E.P.S.

## Competing interests
The authors declare no competing interests.
