## [Peer Review File · Nature Communications]

Primary visual cortex straightens natural video trajectoriesREVIEWER COMMENTS

Reviewer #1 (Remarks to the Author):

This manuscript by Henaff et al explores neural population responses to sequences of natural images, in macaque primary visual cortex. The work builds on work published by several of these authors, presenting the 'temporal straightening hypothesis' (NN, 2019). That paper reported analysis of image sequences (in the pixel domain) and perceptual experiments. This manuscript presents corresponding analysis of neural data. The primary finding is that V1 populations straighten natural image sequences but not artificial ones, and that the straightening is largely due to well-described features of V1 neurons (especially the creation of complex cells).

The work is nicely presented, and very original. The methods are somewhat involved and in places difficult to follow (Point 2 below). There are so many steps, it is a bit difficult to feel fully at ease with the claims. The authors do do a good job of justifying the approach, but it remains a long road from the initial data to the reported outcome. While this may limit the impact of this work, the many methodological innovations, on the other hand, are interesting and will likely help the field beyond the direct application here.

I did struggle a bit with the basic premise and what precisely the authors have shown (Point 1); the paper would be strengthened if these messages were clarified.

1. The premise is that neural circuitry straightens visual input into a representation that better allows prediction based on linear extrapolation. But the test here seems quite different. Specifically, the authors have removed the temporal elements of the sequence by showing discrete movie frames (separated by image blanks). This is mentioned in the discussion but the abstract and introduction should be revised to highlight this limitation more clearly (or a stronger case should be made as to why this limitation can be dismissed). Currently the abstract claims "We test the hypothesis that the visual cortex facilitates predictions...with a straighter temporal trajectory". To my mind, the authors have instead shown that discrete images taken from adjacent frames of a movie elicit neural population responses that change consistently. Indeed, most of the straightening is captured by a simple model that considers just spatial statistics. This is a sensible start but does not really offer a strong test of the hypothesis.
2. One aspect of the analysis was confusing to me: the embedding that takes us from Fig 1d and Fig 1e. Why is the relevant space for evaluating the neural trajectories a neural discriminability space? Please clarify. It might be helpful to add a supplementary figure showing how different sequences of neural responses (like those shown in Fig 1d) map to the discriminability space.
3. The authors report their primary result as a 'relative curvature', comparing the measured responses to a carefully constructed null distribution. They mention in passing that this is because of a bias in the neural curvature estimation that causes the null distribution to deviate from the pixel curvature. It would be helpful to know the magnitude of this bias (i.e. how much difference it would make to use the difference with the null vs the difference with the pixel). In particular, how big is the effect size (a straightening of ~10 deg on average) compared to the bias of the method (null vs pixel).
4. The authors show robust straightening for some sequences and little for others. They offer a qualitative description of some of the apparent differences. It would be useful for the reader to see these differences as well. Please expand Supp Fig 3 to show the sequences (rather than middle frame) for which straightening was strongest/weakest. This might help readers obtain some intuition for the varied outcomes.
5. There are also substantial differences in the straightening afforded by different neural populations. Can this be explained by the presence of the response features discussed in Fig 7 (preponderance of complex cells)? Fig 7e shows the model can explain much of the variance across movies (black points are highly correlated). What about the variance across data sets?

Reviewer #2 (Remarks to the Author):

This manuscript describes analyses indicating that neural responses in the primary visual cortex straighten representations of natural video trajectories. The results are interesting and augment previous recent reports by the authors of this straightening at the level of human perception.

There are several serious concerns about the analyses and experimental set up that need to be addressed. They are listed in order of their importance:

1) The natural stimuli are frames of a natural movies presented in a random order. This is stated so on page 9 in the methods “images are presented randomly over the course of the experiment” On page 4, this choice is motivated as the one that “removes history-dependent mechanisms, such as temporal filtering, motion selectivity, response adaptation and recurrent computation”. However, all of these effects are still present in neural responses and if anything are exaggerated because in addition to being presented in random order, the frames are separated by a blank frame. This increases transients in neural responses.

2) The curvature of neural representations is obtained using a model whose validity is in question (see next point). Only neural populations that are well modeled by it are analyzed and reported. This can bias the results in unpredictable ways.

3) The assumption that spike counts elicited by different images and on different trials are independent of one another is problematic. One could make this assumption but only after including a temporal kernel in the analysis and convolving the sequence of presented frames accordingly

4) The fact that LN-LN fitted to neural responses to noise could only explain responses to natural stimuli are chance levels is very concerning. Most likely this indicates that the stimulus dataset was not large enough to allow for reliable fitting.

5) The authors have not really analyzed straightening depends on the probability of occurrence of the stimulus in the natural world. Are all stimuli taken from a movie deemed to be equally likely to occur in the natural world?

6) The arguments motivating analyses across spatial scales are very confusing. First, one reads that because natural stimuli are scale-invariant the stimuli presented at different scales should elicit the same reduction in curvature. Then, this is followed by results that depend on the spatial scale and the difference in spike rates that they elicit. Perhaps this is due to the limited size of each recorded neural population.

7) It would be useful to see how the curvature of natural images across different spatial and temporal scales, to compare with neural results.

Response to reviewers

We thank the reviewers for their thoughtful feedback. We first address the main points raised by both reviewers, before answering each reviewer point-by-point.

Temporally contiguous vs. randomly presented static frames

We hypothesize that the visual system transforms natural image sequences so as to straighten their temporal trajectories. We previously found perceptual support for this hypothesis, using a series of psychophysical experiments. In that work, as well as in the current work, we chose to focus initially on asking whether the *spatial processing* of the early visual system contributes to these perceptual effects. To assess this, we randomize the natural temporal order of the video frames, removing any potential systematic contribution from motion processing mechanisms to temporal straightening. Of course, as one reviewer remarked, neural responses still peak transiently upon stimulus onset under our stimulus presentation paradigm. But critically, this transient solely depends on the content of the frame being presented, and not on the motion statistics of the natural video. Furthermore, the randomization of frame order ensures that slower history-dependent mechanisms such as response-adaptation will also not give rise to systematic effects in our experiments. We do agree that temporally contiguous stimulus presentation will be interesting to explore, but note in the text that this will require a substantial modification of the experimental protocol. We have adjusted the language in the abstract and introduction to make sure that readers correctly understand the nature of our paradigm from the beginning.

Methodological questions

Both reviewers asked about design decisions in our analysis, and wondered how these decisions affected our results. Specifically, R1 asked how measuring curvature in a “neural embedding space” which accounts for the effects of neural response variability influences our estimates (Q2). R1 also asked how “relative curvature” compares to raw curvature estimates (Q3). R2 asked whether only including neural trajectories that are well fit by our descriptive model influenced our results (Q2). We have now included quantitative evaluation of the impact of each of these choices on our results. We discuss these additional analyses in the point-by-point responses below, and have included several in the supplementary information section of our paper. In brief, none of them affect our findings in a substantial manner.

Reviewer #1 (Remarks to the Author):

This manuscript by Henaff et al explores neural population responses to sequences of natural images, in macaque primary visual cortex. The work builds on work published by several of these authors, presenting the ‘temporal straightening hypothesis’ (NN, 2019). That paper reported analysis of image sequences (in the pixel domain) and perceptual experiments. This manuscript presents corresponding analysis of neural data. The primary finding is that V1 populations straighten natural image sequences but not artificial ones, and that the straightening is largely due to well-described features of V1 neurons (especially the creation of complex cells).

The work is nicely presented, and very original. The methods are somewhat involved and in places difficult to follow (Point 2 below). There are so many steps, it is a bit difficult to feel fully at ease with the claims. The authors do do a good job of justifying the approach, but it remains a long road from the initial data to the reported outcome. While this may limit the impact of this work, the many methodological innovations, on the other hand, are interesting and will likely help the field beyond the direct application here.

I did struggle a bit with the basic premise and what precisely the authors have shown (Point 1); the paper would be strengthened if these messages were clarified.

Reviewer #1, question 1:

The premise is that neural circuitry straightens visual input into a representation that better allows prediction based on linear extrapolation. But the test here seems quite different. Specifically, the authors have removed the temporal elements of the sequence by showing discrete movie frames (separated by image blanks). This is mentioned in the discussion but the abstract and introduction should be revised to highlight this limitation more clearly (or a stronger case should be made as to why this limitation can be dismissed). Currently the abstract claims “We test the hypothesis that the visual cortex facilitates predictions. . . with a straighter temporal trajectory”. To my mind, the authors have instead shown that discrete images taken from adjacent frames of a movie elicit neural population responses that change consistently. Indeed, most of the straightening is captured by a simple model that considers just spatial statistics. This is a sensible start but does not really offer a strong test of the hypothesis.

Author response:

As explained in the section above, we think that our approach represents a valuable first step towards testing the straightening hypothesis, and that contiguous presentation of movies will make for interesting future work. We have adjusted the language in the abstract and introduction to clarify our paradigm from the outset.

Response Figure 1 Comparing the curvature of neural embeddings to that of the associated population firing rate, for the set of V1 trajectories studied in this paper (coarse temporal scale). The curvature of the neural embedding trajectories, used throughout our paper, is plotted on the x-axis. The curvature of the corresponding neural firing rate trajectories is plotted on the y-axis. Both curvature estimates are very similar, implying that response variability is approximately constant over the range of these trajectories. Mean neural curvature, natural sequences: 92.5° (embedding), 92.9° (firing rate); unnatural sequences: 58.6° (embedding), 59.4° (firing rate).

Reviewer #1, question 2:

One aspect of the analysis was confusing to me: the embedding that takes us from Fig 1d and Fig 1e. Why is the relevant space for evaluating the neural trajectories a neural discriminability space? Please clarify. It might be helpful to add a supplementary figure showing how different sequences of neural responses (like those shown in Fig 1d) map to the discriminability space.

Author response:

We consider neural discriminability to be the correct metric for determining neural curvature, as it provides a principled estimate of the ability of downstream circuits to discriminate movie frames, and thus a natural means to express representational distance. Intuitively, two frames that are easy to distinguish perceptually

should be considered far apart, while two frames that can be confused easily should be considered close to each other. The use of discriminability is also consistent with the analysis in our previously published psychophysical measurements. We have added a sentence to the results section that mentions the relation of neural discriminability and perceptual separation.

That said, your comment led us to re-analyze the data, estimating neural curvature in the space of the estimated population firing rate (thus effectively assuming that response variability is approximately constant across movie frames within a single sequence). The results are nearly the same (Response Figure 1). On average, the curvature of natural sequences would be 0.4° higher than those estimated with discriminability, and that of artificial ones 0.8° higher, leaving our conclusions unchanged. We have included this analysis in the Supplementary Information section.

Response Figure 2 Comparison of relative curvature estimates obtained using our debiasing strategy, with biased estimates which compare estimated neural curvature directly to pixel-domain curvature. In order to debias these estimates, we instead compare to the average curvature of a null-model whose true curvature is equal to the pixel-domain curvature, but whose estimates inherit the same bias as the neural curvature estimates. Median relative curvature, natural sequences: -9.9° (debiased), -10.2° (biased); unnatural sequences: 24.4° (debiased), 48.8° (biased).

Reviewer #1, question 3:

The authors report their primary result as a ‘relative curvature’, comparing the measured responses to a carefully constructed null distribution. They mention in passing that this is because of a bias in the neural curvature estimation that causes the null distribution to deviate from the pixel curvature. It would be helpful to know the magnitude of this bias (i.e. how much difference it would make to use the difference with the null vs the difference with the pixel). In particular, how big is the effect size (a straightening of 10 deg on average) compared to the bias of the method (null vs pixel).

Author response:

We compared our unbiased estimates to the biased ones we would have obtained had we compared neural curvature directly to the pixel-domain curvature, rather than that of the null-distribution (Response Figure 2). In this case, the bias in our estimates would only increase the apparent straightening (from a median of 9.9° to 10.2°). Therefore, while we consider that comparing to an explicit null model is the more principled approach, our conclusions are robust to this choice.

For evaluating whether artificial sequences are entangled, the use of a null-model is essential. In this case, curvature of neural sequences can only be larger than that of the pixel-domain sequences (whose curvature is very close to 0), making the null-model the only relevant comparison.

Response Figure 3 Sequences eliciting strongest and weakest straightening (Figure 3C) and entangling effects (Figure 5C). (a) The strongest straightening effect was elicited by a natural sequence of a prairie, primarily consisting of shrubs and grass textures (zoom x 1, movie #20 in Figure 3C). (b) A sequence of a rotating face was the least straightened natural sequence (zoom x 2, movie #1 in Figure 3C). (c) Entangling was weakest for an unnatural sequence of waves (“Carnegie-dam”, zoom x 2, movie #15 in Figure 5C). (d) Entangling was strongest for an unnatural sequence showing bees moving in a hive (zoom x 2, movie #4 in Figure 5C).

Reviewer #1, question 4:

The authors show robust straightening for some sequences and little for others. They offer a qualitative description of some of the apparent differences. It would be useful for the reader to see these differences as well. Please expand Supp Fig 3 to show the sequences (rather than middle frame) for which straightening was strongest/weakest. This might help readers obtain some intuition for the varied outcomes.

Author response:

We have updated Supplementary Figure 3 as suggested (also shown in Response Figure 3).

Reviewer #1, question 5:

There are also substantial differences in the straightening afforded by different neural populations. Can this be explained by the presence of the response features discussed in Fig 7 (preponderance of complex cells)? Fig 7e shows the model can explain much of the variance across movies (black points are highly correlated). What about the variance across data sets?

Author response:

We asked whether the diversity of straightening effects can be explained by the proportion of complex cells in each population. We categorized cells (simple vs complex) by their harmonic responses to their preferred drifting grating ($F1 / F0 < 1 = \text{complex}$). We then compared the proportion of complex cells to relative curvature values (Response Figure 4). We did not find a compelling relationship between the proportion of

complex cells and relative curvature. Although the mechanisms of complex cells are crucial for recapitulating straightening effects in our LN-LN model (Figure 8), the proportion of complex cells within a population does not explain much of the diversity in straightening effects across neural populations.

Regarding explaining the variance across data sets, we have included the following statement in the final paragraph of the “Computational basis of neural straightening”: *While this LN-LN model provides insight into the computational basis of neural straightening, note that it [...] failed to capture cross-population differences in curvature.*

Response Figure 4 Straightening versus proportion of complex cells. Relative curvature (change in curvature) is plotted as a function of the proportion of complex cells in each population. A single red filled circle indicates the mean value across all natural sequences for a given population (total of 7 populations) and the empty red circle above represents the mean value for the same population across unnatural sequences. The correlation coefficient between the proportion of complex cells and straightening effects was near zero ($r=0.06$). Error bars indicate s.e.m across sequences.

Reviewer #2 (Remarks to the Author):

This manuscript describes analyses indicating that neural responses in the primary visual cortex straighten representations of natural video trajectories. The results are interesting and augment previous recent reports by the authors of this straightening at the level of human perception.

There are several serious concerns about the analyses and experimental set up that need to be addressed. They are listed in order of their importance:

Reviewer #2, question 1:

The natural stimuli are frames of a natural movies presented in a random order. This is stated so on page 9 in the methods “images are presented randomly over the course of the experiment” On page 4, this choice is motivated as the one that “removes history-dependent mechanisms, such as temporal filtering, motion selectivity, response adaptation and recurrent computation”. However, all of these effects are still present in neural responses and if anything are exaggerated because in addition to being presented in random order, the frames are separated by a blank frame. This increases transients in neural responses.

Author response:

Please refer to the section “Temporally contiguous vs. randomly presented static frames” above.

Response Figure 5 Comparison of relative curvature of trajectories which meet our inclusion criterion (grey) and those that do not (white). The inclusion criterion only retains trajectories which provide an adequate fit to the observed spike counts, and whose length is above a certain threshold (allowing reliable curvature estimation). Median relative curvature, natural sequences: -9.9° (included), -10.1° (all); unnatural sequences: 24.4° (included), 27.1° (all).

Reviewer #2, question 2:

The curvature of neural representations is obtained using a model whose validity is in question (see next point). Only neural populations that are well modeled by it are analyzed and reported. This can bias the results in unpredictable ways.

Author response:

We consider an inclusion criterion based on model fit to be a principled choice, as model-based curvature estimates should only be considered if the model accurately reflects the data.

Nevertheless, we asked whether our results could be biased by the use of the inclusion criterion (Response Figure 5). We compared the distribution of relative curvature obtained from sequences that met our inclusion criterion, and those that did not. They can largely be described as following the same distribution. Comparing aggregate statistics, the effects we report are, if anything, emphasized if we include all sequences (median relative curvature, natural sequences: -9.9° with included sequences, -10.1° with all sequences; unnatural sequences: $+24.4^\circ$ with included sequences, $+27.1^\circ$ with all). We conclude that our results are not a simple byproduct of the inclusion criterion.

Reviewer #2, question 3:

The assumption that spike counts elicited by different images and on different trials are independent of one another is problematic. One could make this assumption but only after including a temporal kernel in the analysis and convolving the sequence of presented frames accordingly.

Author response:

It is not clear to us how exactly such temporal convolution would inform our description of neural responses. Should the temporal filter take stimulus tuning into account? How exactly? That said, we don't think it is

needed either. The stimulus presentation paradigm was designed to minimize temporal adaptation effects. Specifically, the blank inter stimulus interval allows response rates to return to baseline in between stimulus presentations. An example of this is shown in Response Figure 6. Moreover, the random stimulus order ensures that any remaining adaptation effects cannot impact responses in a systematic manner. We don't consider our treatment of spike counts as being independent and identically distributed across repeated stimulus presentations in this kind of paradigm to be a novel invention. It is standard practice.

Response Figure 6 Peri-stimulus time histograms (PSTH) from an example V1 cell. We alternated presentation of static image frames (on for 200 ms, indicated by grey background) with a blank screen (on for 110 ms). Each frame was repeated 50 times over the course of the experiment. For every trial, we computed spike rates using 10 ms wide non-overlapping time bins. The thick red line illustrates the mean firing rate, the red shaded regions indicate the associated s.e.m. (a) PSTHs averaged across all frames from an example sequence (movie 10, “leaves-wind”). The top row shows responses to the natural version of the movie, the bottom row shows responses to the unnatural version. The left column shows responses to zoom x 1, the right column to zoom x 2. (b) Same plotting convention as in (a). PSTHs averaged across a single frame of the example sequence.

Reviewer #2, question 4:

The fact that LN-LN fitted to neural responses to noise could only explain responses to natural stimuli are chance levels is very concerning. Most likely this indicates that the stimulus dataset was not large enough to allow for reliable fitting.

Author response:

The transition from noise stimuli and gratings to natural images represents a considerable domain gap. We don't find it surprising that a simple LN-LN model fails to predict responses of cortical cells to natural images. Much earlier in the visual pathway, this domain gap has been shown to result in similar model failures. For example, a pseudo-linear model (the Generalized Linear Model) of retinal ganglion cells can capture the responses elicited by white noise stimuli well, but fails to generalize to responses elicited by natural images (Heitman, Brackbill, Greshner, Sher, Litke, and Chichilnisky, 2016 – Testing pseudo-linear models of responses to natural stimuli in the primate retina). We find it therefore all the more surprising and interesting that as simple a model as our LN-LN model is able to capture some of the straightening behavior of cortical populations.

Reviewer #2, question 5:

The authors have not really analyzed straightening depends on the probability of occurrence of the stimulus in the natural world. Are all stimuli taken from a movie deemed to be equally likely to occur in the natural world?

Author response:

We agree that a full assessment of the dependency of straightening on the probability of stimulus occurrence would require an accurate probabilistic model of natural image sequences. This represents an open research question. In this paper, our logic relies on the assumption that the natural videos we selected are sufficiently likely that the visual system should seek to straighten them, and that this is not the case for the artificial videos we created. We furthermore assume that the spatial and temporal scale manipulations are mild enough to preserve the relative likelihood across movies.

Reviewer #2, question 6:

The arguments motivating analyses across spatial scales are very confusing. First, one reads that because natural stimuli are scale-invariant the stimuli presented at different scales should elicit the same reduction in curvature. Then, this is followed by results that depend on the spatial scale and the difference in spike rates that they elicit. Perhaps this is due to the limited size of each recorded neural population.

Response Figure 7 Curvature values as a function of average firing rate for all natural movies, across two spatial scales. We dissected Figure 6C's 'relative curvature' into its two elements: (a) neural - pixel curvature, and (b) null model - pixel curvature, as a function of spike rates (i.e. the difference between (a) and (b) produces Figure 6C). Each point illustrates a dataset, following the same color code as in Figure 6C (red: zoom x 1, grey: zoom x 2). Regression lines are plotted in bold lines and annotated with Pearson's correlation coefficients.

Author response:

We revisited this analysis and arrived at a slightly modified interpretation that is consistent with the reviewer's suggestion. As we had reported previously, higher firing rates are associated with stronger straightening effects at both spatial scales. Why is this so? Is it simply an effect of SNR (as we suggested previously)? We think not. Comparing the expected curvature under the null-model to the pixel-domain value reveals whether curvature estimates happen to be biased for a given data-set. However, this statistic exhibits no systematic dependency on firing rate (Response Figure 7, panel b). The systematic increase in straightening with firing rate (Response figure 7, panel a) therefore is a real physiological effect. Why do better driven populations straighten natural movies more strongly? We don't really know. One possibility is that movies that contain cells' preferred features engage non-linearities of V1 circuits in a unique manner, and that these non-linearities in turn act to straighten the temporal response trajectory. There is some evidence for the existence of such specialized non-linear mechanisms in complex cells, but not in simple cells (Felsen, Touryan, Han, and Dan, 2005 – PLoS Biology: Contrast-sensitivity to Visual Features in Natural Scenes). This is the interpretation we now suggest in the paper. As a consequence, we think the reviewer is correct in suggesting that population size

may be an important factor here. If we could record from substantially larger populations, then they would presumably be diverse enough to contain a subset of neurons well-tuned to each image sequence.

Reviewer #2, question 7:

It would be useful to see how the curvature of natural images across different spatial and temporal scales, to compare with neural results

Author response:

In Response Figure 8 we compare the pixel-domain curvature across spatial and temporal scales. Consistently with the argument that the statistics of natural image sequences are invariant for the scales we consider, we find their curvature to be highly preserved across both spatial ($r = 0.96$) and temporal ($r = 0.91$) scales. The pixel-domain curvature of unnatural sequences is zero by construction, with only slight deviations due to the quantization of individual frames required for presentation on a monitor. These deviations are also highly consistent across scales.

Response Figure 8 Comparison of pixel-domain curvature across spatial and temporal scales. We use Spearman correlation to quantify the relationship across scales.

REVIEWER COMMENTS

Reviewer #1 (Remarks to the Author):

The authors have addressed the majority of my concerns. I support publication.

Reviewer #2 (Remarks to the Author):

I remain very concerned about experimental design. To summarize, the video frames are shown in random order with blank frames in between. The resultant responses are then re-ordered to match the original temporal sequence. These are all very artificial manipulations that destroy the natural dynamics of visual processing. In addition, the authors acknowledge that the use of blanks generated transients. They state that “transient solely depends on the content of the frame being presented”, when in fact it will be mostly driven by the contrast change between the movie frame and the blank, and only weakly depend on a particular spatial content. Thus, we are mostly studying regular transients in neural responses stitched at regular intervals. Why should this be of interest? The motivation for doing this is “These manipulations removed any systematic contributions from fast motion-selective mechanisms, as well as from history-dependent mechanisms such as response adaptation”. However, these adaptive mechanisms contribute have been shown many times in the past to increase the efficiency of neural representations.

In addition, because the straightening of trajectories is directly related to the evoked firing rate (Fig. 6c), then this can explain why unnatural sequences (which evoke smaller firing rates) are made more curved by neural responses.

These concerns combined with the relative smallness in the reported effects (straightening is not observed for all neural populations or sequences) reduces the impact of the study.

We thank the reviewers for their comments. Please find our point-by-point reply below.

** Reviewer #1 (Remarks to the Author):*

The authors have addressed the majority of my concerns. I support publication.

We thank the reviewer.

** Reviewer #2 (Remarks to the Author):*

I remain very concerned about experimental design. To summarize, the video frames are shown in random order with blank frames in between. The resultant responses are then re-ordered to match the original temporal sequence. These are all very artificial manipulations that destroy the natural dynamics of visual processing. In addition, the authors acknowledge that the use of blanks generated transients. They state that “transient solely depends on the content of the frame being presented”, when in fact it will be mostly driven by the contrast change between the movie frame and the blank, and only weakly depend on a particular spatial content. Thus, we are mostly studying regular transients in neural responses stitched at regular intervals. Why should this be of interest? The motivation for doing this is “These manipulations removed any systematic contributions from fast motion-selective mechanisms, as well as from history-dependent mechanisms such as response adaptation”.

However, these adaptive mechanisms contribute have been shown many times in the past to increase the efficiency of neural representations.

The goal of this work is to study the neural correlate of a perceptual effect (Temporal straightening of natural image sequences, as documented in Hénaff et al. (2019) – *Nature Neuroscience*). It requires an experimental design that reasonably approximates the stimulus conditions of the perceptual experiment, and thus the use of static images. This goal is explicitly stated in the introduction. To further clarify this, we have added the following statement to the results:

In our previous perceptual experiments, we used an experimental protocol that relies on the presentation of static images, thereby isolating the contribution of spatial processing mechanisms. Here, we seek to study the neural basis of these perceptual effects. We therefore used the same stimulus presentation method.

We agree with the reviewer that natural stimulus dynamics will affect the neural representation and will likely change neural temporal trajectories. We fully agree that this is a question that deserves further study. To further clarify this, we have added the following statement to the discussion (new text marked **in red**):

...How would these results generalize to representations of continuous streams of images? If temporal straightening is a prominent goal of visual processing, we might expect stronger effects under more natural stimulus conditions. Indeed, the inertia of physical systems makes their three-dimensional physical trajectories locally linear, and thus predictable. **Although linear motion trajectories display a stable distribution of spatiotemporal energy, they can yield highly curved image-domain trajectories, since the brightness of individual pixels can change abruptly due to the passage of sharp edges, sudden occlusion, or changes in illumination. Engaging motion-selective neural mechanisms may therefore further increase temporal straightening. This is an important question for future work.**

In addition, because the straightening of trajectories is directly related to the evoked firing rate (Fig. 6c), then this can explain why unnatural sequences (which evoke smaller firing rates) are made more curved by neural responses.

Recall that we use a null-model to derive a baseline expectation given the observed firing rate statistics. Our analysis shows entangling much beyond this expectation. Firing rate as such can thus not “explain” these results.

These concerns combined with the relative smallness in the reported effects (straightening is not observed for all neural populations or sequences) reduces the impact of the study.

We respectfully disagree with this statement. The effects are diverse, but systematic (as clearly revealed by our statistical tests). We think their magnitude is consistent with the hypothesis that temporal straightening arises from a cascaded computation and gradually increases in magnitude along the visual hierarchy. We consider this an important question for future work, and we say so explicitly in the discussion.